# Circularly polarized OLEDs from chiral plasmonic nanoparticle-molecule hybrids

Jiapeng Zheng[1,2], Yuang Fu[2], Jing Wang[3], Wei Zhang[4], Xinhui Lu[2], Hai-Qing Lin[5], Lei Shao[3] ✉ & Jianfang Wang[2] ✉

Organic light-emitting diodes (OLEDs) supporting the direct emission of circularly polarized (CP) light are essential for numerous technologies. The realization of CP-OLEDs with large dissymmetry ($g_{EL}$) factors and high external quantum efficiencies (EQEs) has been accepted as a considerable challenge. Here we demonstrate the realization of efficient CP-OLEDs based on the assembly of chiral plasmonic nanoparticles (NPs) and supramolecular aggregates. The chiral plasmonic NPs serve as the chiral scaffold and chiral optical nanoantenna to modulate the circularly polarized absorption and emission of the supramolecular chromophores. We employ different chiral plasmonic NPs to construct various CP-OLEDs with the emission dominated by chiral excitons or chiral plasmons. The CP-OLED showing a high EQE of 2.5% and a large $g_{EL}$ factor of 0.31 is achieved, as a result of multiscale chirality transfer, plasmonic enhancement, and the suppression of the overshoot effect. The proposed schemes are compatible with the current manufacturing technology of OLEDs. This work demonstrates that chiral plasmonic NPs can be promising candidates in chiral photoelectric devices.

Circularly polarized light sources play vital roles in applications of three-dimensional display and imaging[1], enantiomeric analysis[2], asymmetric synthesis[3], and quantum teleportation[4]. A linear polarizer and a quarter wave plate are commonly used to obtain CP light, leading to large optical power loss and requiring additional optics. CP-OLEDs are therefore commercially interesting because of their light weight, solution processability, and potential to construct flexible devices with low-power consumption[5–7]. The EQE and $|g_{EL}|$ factor are the main parameters for characterizing the performances of CP-OLEDs. The former represents the generation efficiency of emitted photons, while the latter shows the degree of difference in the numbers of left-handed circularly polarized (LCP) and right-handed circularly polarized (RCP) photons emitted. Traditional CP-OLEDs fabricated by various chiral organic emitters have been revealed to show an inverse relationship

between the EQE and $|g_{EL}|$ factor[7]. For example, chiral heterometallic clusters have been implemented in CP-OLEDs[5] with EQEs close to 21%, but their $|g_{EL}|$ factors are limited at the level of $10^{-4}$. Lanthanide complexes can achieve an extremely high $|g_{EL}|$ of 0.5–1.0, but their EQEs are very low (0.05%–0.48%)[8,9]. The improved chiral organic emitters are therefore needed to achieve significant improvements both in EQE and $|g_{EL}|$.

Chiral complex composites, prepared by the integration of organic emitters with chiral nanomaterials, are a promising class of chiral-emitting materials[10–13]. In these composites, chiral nanomaterials such as twisted substrates and chiral metasurfaces can serve as chiral templates or chiral-transfer media to guide the chiral emissions of achiral emitters with high quantum yields[14,15]. These composites show broad applicability in the incorporation of various emissive materials

[1]School of Artificial Intelligence Science and Technology, Institute of Photonic Chips, University of Shanghai for Science and Technology, Shanghai 200093, China. [2]Department of Physics, The Chinese University of Hong Kong, Shatin, Hong Kong SAR 999077, China. [3]State Key Laboratory of Optoelectronic Materials and Technologies, Guangdong Provincial Key Laboratory of Display Materials and Technologies, School of Electronics and Information Technology, Sun Yat-sen University, Guangzhou, Guangdong 510275, China. [4]National Key Laboratory of Computational Physics, Institute of Applied Physics and Computational Mathematics, Beijing 100088, China. [5]School of Physics, Zhejiang University, Hangzhou, Zhejiang 310058, China. ✉e-mail: shaolei5@mail.sysu.edu.cn; jfwang@phy.cuhk.edu.hk

and are accepted as alternatives to conventional chiral emitters. However, most of chiral complex composites are made by non-conductive materials or fabricated with limited scales[14,15]. Emissions with both high quantum yields and strong CP anisotropy were usually achieved through optical excitation. The design of chiral complex composites with improved chiroptical properties and good electrical properties is accepted as the prerequisite for developing the next-generation CP electroluminescence (CPEL) devices.

In this work, we overcome these urgent challenges and develop three types of CP-OLEDs with large light-emitting areas, low driving voltages, and efficient CP emissions. The chiral complex composites were fabricated through the integration of synthesized chiral plasmonic NPs and supramolecular aggregates (Fig. 1). A reported halide-assisted differential growth strategy was adopted to synthesize these chiral NPs with continuously tunable chiroptical properties (Supplementary Fig. 1 and "Methods")[16]. The emerging chiral Au NPs with helicoidal morphologies, such as 432 helicoid III, 432 helicoid IV, and nanotriskelions, have shown enhanced chiral light–matter interactions (Fig. 1a–c)[16–18]. The 432 helicoid III NPs and Au nanotriskelions show strong chiroptical responses, coming from their chiral arms with fourfold rotational symmetry along the <100> directions and threefold rotational symmetry along the <111> directions, respectively (Fig. 1b, c). The 432 helicoid IV NPs' chiral arms exhibit opposite chirality along the <100> and <111> directions, leading to a weak far-field chiroptical response (Fig. 1a, Supplementary Figs. 2, 3). We used 5,6-dichloro-2-[[5,6-dichloro-1-ethyl-3-(4-sulfobutyl)-benzimidazol-2-ylidene]-propenyl]−1-ethyl-3-(4-sulfobutyl)-benzimidazolium hydroxide (TDBC) as the chromophore molecule (Fig. 1d). We focused on two critical mechanisms (Fig. 1e,f) and designed various CP-OLED architectures (Fig. 1g, h). The first is the transfer of the geometric chirality from NPs to molecule aggregates[19]. Chiral NPs can direct the chiral stacking of supramolecular aggregates to generate chiral Frenkel excitons. This chiral-exciton-dominated CP-OLED-1 device was fabricated based on 432 helicoid IV−molecule hybrids, showing the bisignate CPEL with $|g_{EL}|$ of ≈ 0.15 and EQE of 1.3% (Fig. 1e). The second is the transfer of optical chirality[20]. The strong chiroptical response of the plasmonic 432 helicoid III and nanotriskelion NPs enables the circular-polarization filtering of unpolarized emissions through circular-polarization-preferential scattering and absorption by the chiral NPs (Fig. 1f). This chiral-plasmon-dominated CP-OLED-2 device was fabricated based on 432 helicoid III−molecule hybrids, showing a higher EQE value (2.5%) and a larger $|g_{EL}|$ factor (0.31). Furthermore, we developed the CP-OLED-3 device by separating the nanotriskelions and molecule composites, exhibiting extraordinary CPEL with EQE of 2.1% and $|g_{EL}|$ of ≈ 0.32 (Fig. 1g, h).

## Results

### Fabrication of chiral plasmonic NP−molecule hybrids

We first prepared TDBC-based layer-by-layer J-aggregate films that support delocalized Frenkel excitons through evaporation-driven self-assembly (Supplementary Figs. 4, 5)[21,22]. Different from commonly used fluorescent dyes such as Rhodamine (Supplementary Fig. 6),

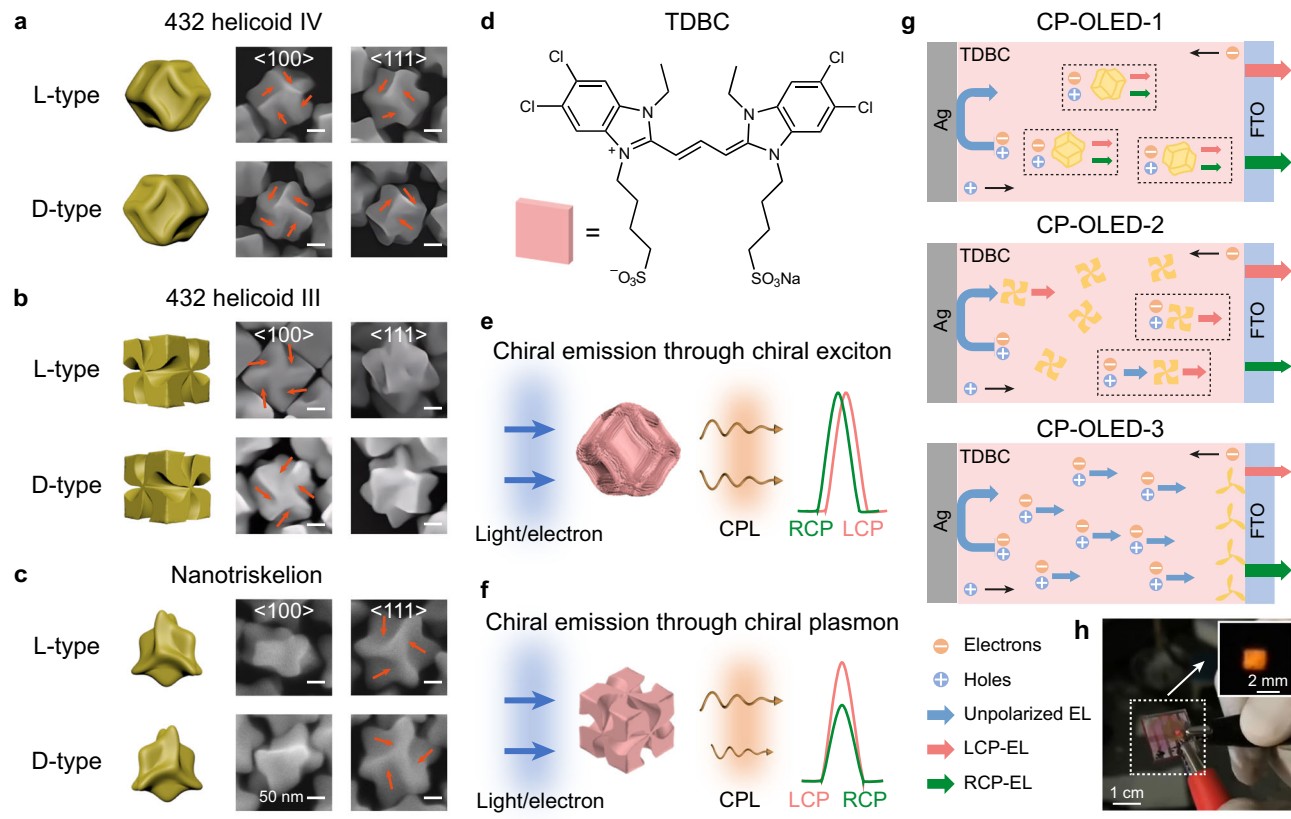

**Fig. 1 | Chiral plasmonic NPs in the emissive layer of CP-OLEDs. a–c** Schematics and scanning electron microscopy (SEM) images of the chiral plasmonic NPs. The 432 helicoid IV NPs (**a**) show rhombic-dodecahedral shapes. The 432 helicoid III NPs (**b**) are of a cubic geometry, with the six faces exhibiting pinwheel-like patterns. The nanotriskelions (**c**) exhibit the anisotropic geometric morphology with triskelion-shaped wrinkles on the top and bottom surfaces. The L- and D-type chiral NPs were obtained from L- and D-GSH, respectively. The distorted edges of the chiral NPs are shown by the orange arrows. All scale bars share the same value. **d** TDBC monomer, schematically represented by a pink platelet. **e, f** Schematics illustrating the chirality transfer from the chiral plasmonic NPs to the emission of the CP-OLEDs, generating circularly polarized luminescence. The chiral emissions result from either the formation of chiral excitons (**e**) or the modulation by chiral plasmon resonance (**f**). **g** Schematic diagrams illustrating the device structures of the CP-OLEDs in this work. **h** Photographs showing a typical CP-OLED device and its electroluminescence (inset).

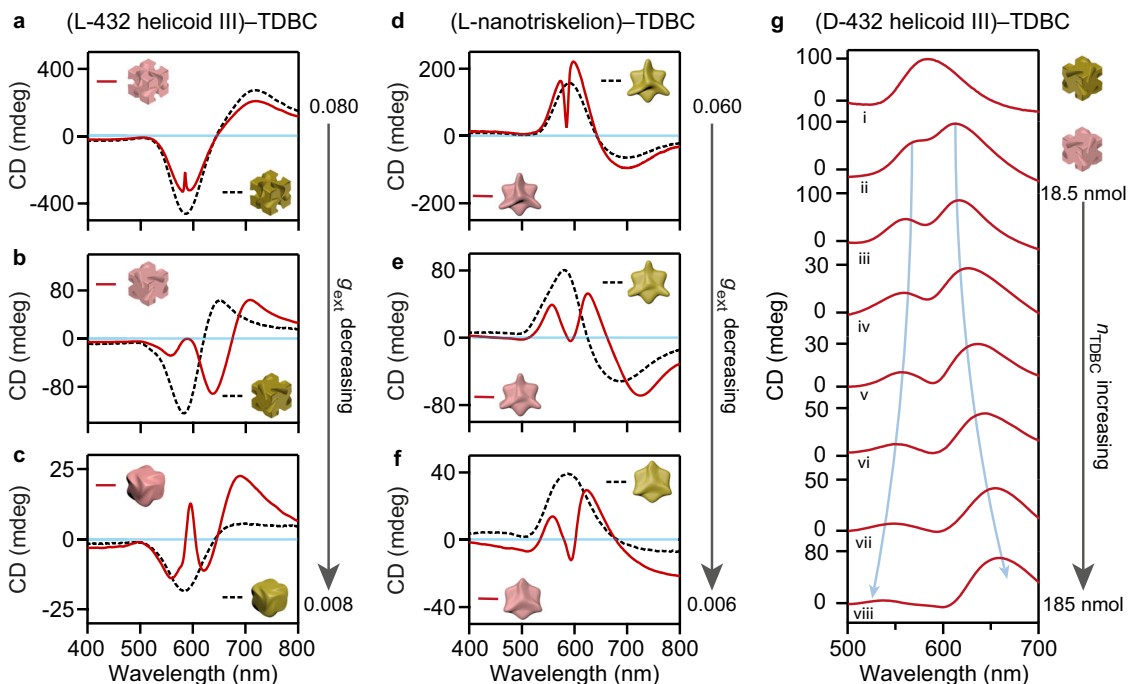

**Fig. 2 | Chiral plasmon–exciton interaction in the hybrids between the chiral plasmonic NPs with strong geometrical chirality and TDBC aggregates.**
**a–f** Extinction CD spectra of the chiral plasmonic NPs in aqueous solutions (black dashed lines) and corresponding hybrid films prepared by forming TDBC aggregates on the surfaces of the chiral plasmonic NPs (red lines). The hybrids were fabricated by the same amount of the chiral NPs (50 μL) and TDBC (0.92 mM, 80 μL). **g** Chiral-plasmon–chiral-exciton interaction evolving from the weak- to strong-coupling-like regime as the amount of TDBC ($n_{TDBC}$) was increased in the (D-432 helicoid III NP)–(TDBC aggregate) hybrid films. $n_{TDBC}$ = 0, 18.5, 37, 55.5, 74, 111, 150 and 185 nmol for lines i–viii, respectively. The blue arrows show the enlarged CD splitting signals with the increased $n_{TDBC}$. The SEM images and $g_{ext}$ spectra of the chiral NPs are shown in Supplementary Fig. 7. Source data are provided as a Source Data file.

achiral TDBC monomers can self-assemble into aggregates with random chiral configurations under solvent evaporation[23,24], endowing the aggregate film with a very weak ensemble chiroptical response. The formed chiral J-aggregate films exhibit a narrow absorption band at ≈ 585 nm, an emission peak at ≈ 600 nm, a bisignate absorption circular dichroism (CD) signal centered around the absorption band, and a Davydov splitting of 68 meV (Supplementary Fig. 4). The observed bisignate CD spectrum indicates the energy splitting between Frenkel excitons owing to interchromophoric interactions[25,26]. Previous studies have demonstrated that various (plasmonic NP)–aggregate hybrids supporting enhanced light–matter interactions, such as (achiral Au nanorod)–(chiral aggregate)[27,28] and (chiral Au nanorod dimer)–(TDBC aggregate) systems[29], can modify the chiroptical properties of the hybrid systems, resulting in mode splitting in CD spectra. We hereby demonstrate that the coupling between the chiral NPs and chiral TDBC aggregates can result in richer phenomena, including enlarged CD splitting and enhanced bisignate CD signals. Chiral plasmonic NP–molecule hybrids were fabricated through the assembly of TDBC aggregates and the colloidal chiral plasmonic NPs. The chiral NPs were designed (1) to possess helicoidal morphologies causing the formation of TDBC aggregates with certain long-range geometric chirality[25], and (2) to exhibit chiroptical responses spectrally overlapping the absorption CD band of the TDBC aggregates. The chiroptical responses of the chiral NPs were characterized by extinction dissymmetry factors $g_{ext}$, with $g_{ext} = 2(A_{LCP} - A_{RCP})/(A_{LCP} + A_{RCP})$, where $A_{LCP}$ and $A_{RCP}$ are the extinction spectra for LCP and RCP incident light. The resulting NP–molecule hybrids support the generation of chiral plasmons and chiral excitons, whose interaction was systematically investigated by CD and circularly polarized photoluminescence (CPPL) measurements. We observed the CD, CPPL, and CPEL spectral evolution of the hybrids from the chiral-plasmon-like to chiral-exciton-like feature with decreasing $g_{ext}$ factors.

## Enlarged CD splitting in 432 helicoid III–molecule hybrids

We first synthesized 432 helicoid III and nanotriskelion NPs with controllable helicoidal structures. The opposite geometric chiralities between L-432 helicoid III and L-nanotriskelions (Fig. 1b, c) result from the halide-assisted favored growth along different crystal axes in the presence of the same L-type chiral ligand[16]. The two types of NPs therefore present opposite chiroptical responses (Supplementary Figs. 2, 3). When these chiral NPs with large $g_{ext}$ were hybridized with TDBC aggregates, a small splitting appeared around the exciton energy in the extinction CD spectra, indicating that the chiroptical response of the hybrid film is dominated by chiral plasmons (Fig. 2a–f). The splitting became more distinct when the plasmonic NPs were synthetically designed to have weaker geometric chiral dissymmetry (Supplementary Fig. 7) and thus smaller $|g_{ext}|$ (Fig. 2a–f). The CD sign at the splitting peak (dip) could even be reversed under certain conditions, in agreement with our theoretical analysis (Supplementary Fig. 8). Given that the absorption CD of the TDBC aggregates prepared in the absence of chiral NPs (Supplementary Fig. 4e) was very weak, the splitting in the extinction CD spectra of the hybrid films indicates strong interactions between the chiral plasmonic NPs and the TDBC aggregates.

To ascertain how the chiral NPs affect the chiroptical properties of molecule aggregates, we prepared hybrid films by employing TDBC monomers with varying concentrations. The D-432 helicoid III NPs with $g_{ext}$ = 0.02 at 585 nm were selected as the scaffold for molecular assembly since their $g_{ext}$ peak matches the absorption band (585 nm) of the TDBC J-aggregates (Supplementary Fig. 7). Gradually enlarged energy splitting from 20 to 430 meV was observed in the extinction CD spectra as the amount of TDBC ($n_{TDBC}$) in the film was increased, showing the evolution of the (chiral plasmon)–(chiral exciton) interaction from the weak- to strong-coupling-like regime (Fig. 2g). The energy splitting was further found to scale with the square root of

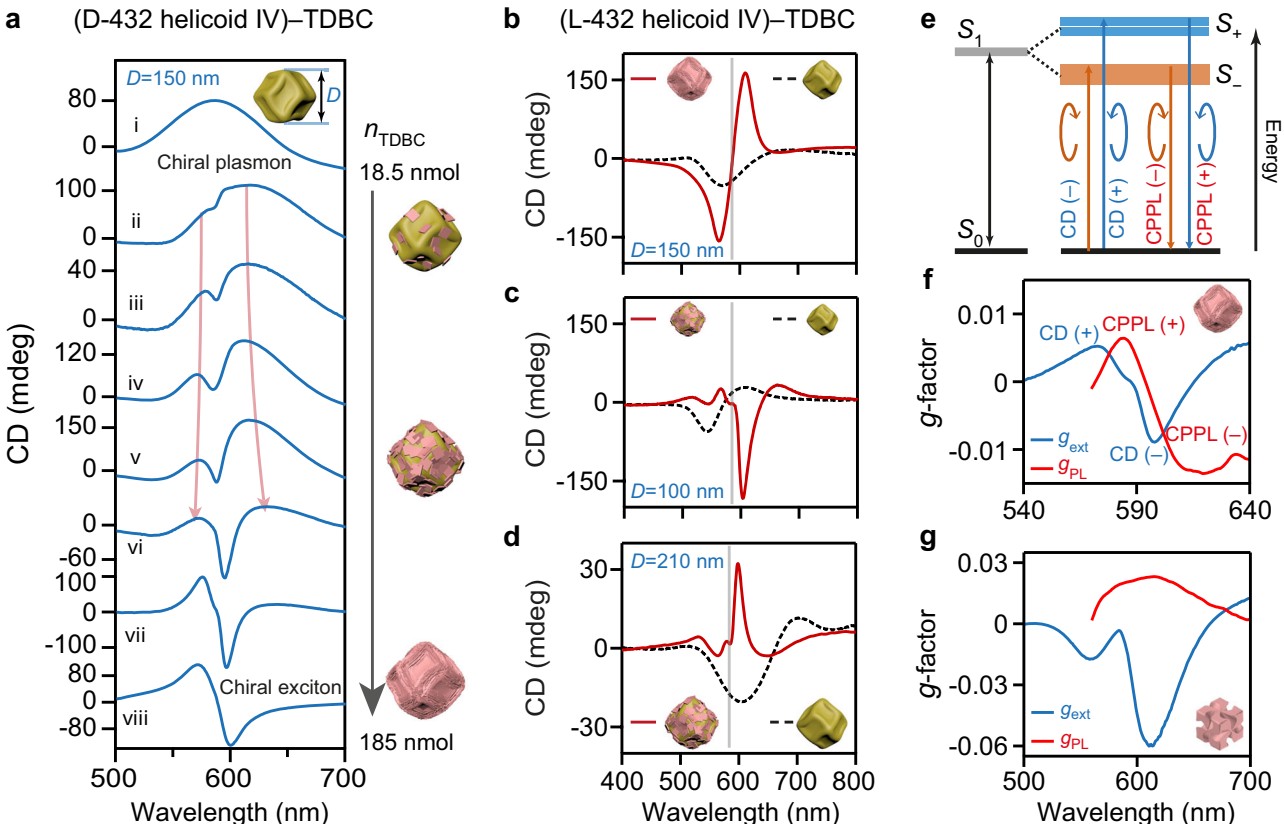

**Fig. 3 | Formation of chiral Frenkel excitons in the hybrids of the chiral plasmonic 432 helicoid IV NPs and TDBC aggregates. a** Extinction CD spectra of the (D-432 helicoid IV NP)–(TDBC aggregate) hybrid films varying from the chiral-plasmon- to chiral-exciton-like feature as the initial $n_{TDBC}$ was increased during the hybrid film preparation. The size of the chiral NPs is $D = 150$ nm. $n_{TDBC} = 0$, 18.5, 37, 55.5, 74, 111, 150 and 185 nmol for lines i–viii, respectively. The red arrows show the enlarged CD splitting signals with the increased TDBC amount. **b–d** Extinction CD spectra of the (differently sized L-432 helicoid IV NP)–(TDBC aggregate) hybrid films exhibiting different features. The gray lines show the extinction band of TDBC. The hybrid film composed of the NPs with $D = 150$ nm and high $n_{TDBC}$

exhibits the chiral-exciton-like feature (**b**). The hybrid films composed of the NPs with $D = 100$ nm (**c**) and $D = 210$ nm (**d**) and moderate $n_{TDBC}$ exhibit the chiral plasmon–exciton coupling feature. **e** Energy diagram of the chiral excitons in TDBC J-aggregates derived from the CD and CPPL spectral measurements. **f** Extinction dissymmetry factor spectrum (blue line) and fluorescent $g_{PL}$ spectrum (red line) of the (D-432 helicoid IV NP)–(TDBC aggregate) hybrid film. **g** Extinction dissymmetry factor spectrum (blue line) and fluorescent $g_{PL}$ spectrum (red line) of the (L-432 helicoid III NP)–(TDBC aggregate) hybrid film. The SEM images and $g_{ext}$ spectra of the 432 helicoid IV NPs were shown in Supplementary Fig. 12. Source data are provided as a Source Data file.

$n_{TDBC}$ (Supplementary Fig. 8)[30,31]. Our observation suggests that the higher $n_{TDBC}$ gives rise to an increase in the number of chiral excitons in the TDBC aggregates formed on the chiral NP surface. The energy splitting in the (432 helicoid III NP)–(TDBC aggregate) hybrid film is drastically larger than that observed in (chiral Au nanorod)–(TDBC aggregate) hybrids in ref. 32. It is also larger than other chiral plasmon–exciton systems that we prepared from the same chiral NPs, including the mixture solution of the NPs and TDBC (Supplementary Fig. 9), NP–Rhodamine 590 (R590) (Supplementary Fig. 10), and NP–Rhodamine 640 (R640) hybrid films (Supplementary Fig. 11). Chiral molecule aggregates are absent in all the Rhodamine samples. We therefore hypothesize that the largest energy splitting in the (432 helicoid III NP)–(TDBC aggregate) hybrids results from the enhanced optical asymmetry in the TDBC aggregates that are stacked on the NP surface in a long-range chiral configuration, as well as energy matching between the chiral plasmon resonance and the exciton absorption band.

### Bisignate CD signals in 432 helicoid IV–molecule hybrids
Previous studies suggested that the supramolecular chirality of molecular J-aggregates can be tailored by the supporting metal surface[28]. To verify our hypothesis that the chiral NPs are able to direct the chiral stacking of TDBC aggregates and therefore give rise to excitons with

enlarged chiroptical responses, we further prepared chiral plasmonic–excitonic hybrids from the Au 432 helicoid IV NPs. In contrast to the 432 helicoid III and nanotriskelion NPs, the 432 helicoid IV NPs exhibit opposite structural chirality along the NP <111> and <100> directions (Fig. 1a, Supplementary Fig. 12) and thus a very weak far-field chiroptical response (Supplementary Fig. 3)[16]. We thereafter prepared the hybrid films by employing TDBC monomers with different amounts. The 432 helicoid IV NPs of $D = 150$ nm in size with their chiral plasmon resonance matching the TDBC J-aggregate absorption band were chosen (Fig. 3a and Supplementary Fig. 12). As $n_{TDBC}$ was increased, the hybrid extinction CD spectrum evolved from a chiral-plasmon-like to chiral-exciton-like feature. We further performed Kramers-Kronig transformation of the CD spectra to obtain the corresponding optical rotatory dispersion (ORD) spectra[33], which exhibit a narrow peak at the exciton absorption energy. The ORD peak became stronger at higher $n_{TDBC}$ values (Supplementary Fig. 13), indicating the existence and gradual enrichment of chiral excitons. In addition, the hybrids composed of D-432 helicoid IV and its L-type counterpart displayed mirror-symmetrical extinction CD and ORD spectra with each other, demonstrating the correspondence between the supramolecular chirality of the formed TDBC aggregates and the structural chirality of the chiral NPs (Supplementary Fig. 14, also see Fig. 3a(viii) and Fig. 3b). Control experiments on (Au nanosphere)–(TDBC

aggregate) hybrid films showed very weak extinction CD signals, with the ORD peak intensities being one to two orders of magnitude smaller than those of the (432 helicoid IV NP)–(TDBC aggregate) hybrids (Supplementary Fig. 15). Therefore, the plasmonic near-field enhancement alone cannot lead to a large exitonic absorption chirality from supramolecular chromophores. Our results strongly support the argument that the chiral NPs can direct the chiral stacking of TDBC aggregates, and as a result generate chiral excitons in the formed supramolecular chromophores. Chirality transfers from the chiral NPs to the molecule aggregates coated on them, similar to the way that twisted elastic substrates transfer chirality to conformally coated NP chains[19].

The extinction CD peak or dip splitting owing to chiral plasmon–exciton coupling was also observed in the (432 helicoid IV NP)–(TDBC aggregate) hybrid films as long as neither the chiral plasmons nor the chiral excitons dominate the chiroptical response of the hybrid (see Fig. 3c, d). The splitting produced a dip or peak near the TDBC J-aggregate absorption band, depending on the sign of the NP extinction CD at the molecular exciton resonance wavelength. We observed a sign inversion in the extinction CD spectra of the hybrids composed of the 432 helicoid IV NPs with $D = 100$ and 210 nm, suggesting that both of the NP samples support strong chiral plasmon–exciton interaction.

Besides the chiroptical response of the plasmonic NPs, the molecular exciton properties can also largely modulate the chiral plasmon–exciton interaction. Such an exciton-dependent behavior can be applied for molecular sensing, as we demonstrated in employing the variations in the extinction CD spectral splitting of the hybrids to quantify the molecule concentration (Supplementary Fig. 16). The interesting chiral plasmon–exciton coupling may lead to potential applications in the enantioselectivity sensing and configuration analysis of supramolecular systems, while the detailed coupling mechanism requires further study.

## Chiral-exciton-dominated CPPL

Because of the correlation between the absorption CD and circularly polarized luminescence (CPL) emission of chiral molecules (Fig. 3e)[25,26], the (chiral plasmonic NP)–molecule hybrids are promising for developing emitters with improved CPL performance. The CPL performance of the emitters can be characterized by both CPPL and CPEL measurements. We first examined the CPPL responses of the (chiral plasmonic NP)–molecule hybrid films. The CPPL could result from the following two mechanisms. (1) Chiral Frenkel excitons in the J-aggregates were generated, emitting photons with opposite helicities from two chiral excited states $S_+$ and $S_-$. (2) The emitted photons were filtered through circular-polarization-preferential scattering and absorption by the chiral NPs. To disentangle the contributions to the CPPL from the chiral excitons and from the chiral plasmonic nanoantenna, we again chose the 150-nm 432 helicoid IV NPs with the weakest far-field chiroptical response as the plasmonic component and prepared hybrid films by employing various chromophore molecules including R590, TDBC, and R640. The hybrid films exhibit strong PL with emission peaks positioned at 580, 600, and 615 nm, respectively. The CPPL dissymmetry factor $g_{PL} = 2(I^P_{LCP} - I^P_{RCP})/(I^P_{LCP} + I^P_{RCP})$ was measured to evaluate the circular polarization degree of the PL of the hybrids, where $I^P_{LCP}$ and $I^P_{RCP}$ are the LCP and RCP emission intensities under the unpolarized excitation. For the (432 helicoid IV NP)–(TDBC aggregate) hybrid, $g_{PL}$ followed the spectral shape of the hybrid extinction CD $g_{ext}$, exhibiting a redshifted, bisignate profile (Fig. 3f). Specifically, when the D-432 helicoid IV NP with $g_{ext} = 0.006$ at 580 nm was employed, the hybrid containing TDBC aggregates presented a differential CPPL peak value one order of magnitude larger than those of the hybrids containing R590 and R640 (Fig. 3f, Supplementary Fig. 13). The weaker CPPL responses of the (432 helicoid IV NP)–Rhodamine hybrid films suggest that the plasmon resonance of

the 432 helicoid IV NP does not provide a strong chiral optical near-field that can modulate the CPPL at 580–615 nm through the chiral Purcell effect[34,35]. The formation of chiral excitons therefore dominates in contributing to the CPPL of the (D-432 helicoid IV NP)–(TDBC aggregate) hybrid film, whose $g_{PL}$ reached −0.014 at 620 nm and 0.007 at 585 nm (Fig. 3f). The dominance of chiral excitons was further confirmed by the TDBC-concentration-dependent CPPL experiments (Supplementary Fig. 13). The hybrids constructed from the L- and D-432 helicoid IV NPs presented mirror-symmetric CD, ORD, and CPPL spectra (Supplementary Fig. 14). In comparison, the control sample of the (Au nanosphere)–(TDBC aggregate) hybrid film also presented the chiral emissions of Frenkel excitons, but the $|g_{PL}|$ is 10 times smaller (Supplementary Fig. 15e). Such results further evidenced the dominant contribution to the CPPL from chiral excitons in the (432 helicoid IV NP)–(TDBC aggregate) hybrid and the important role of the chiral NPs in directing the chiral assembly of TDBC aggregates on their surface[36]. The resultant large number of chiral excitons makes the (chiral plasmonic NP)–molecule hybrids a valuable candidate for multifunctional chiral optoelectronic devices such as chiral emitters and chiral photodiodes[37].

The bisignated CPPL signal reveals an anti-Kasha emissive nature of the (432 helicoid IV NP)–(TDBC aggregate) hybrid. The same anti-Kasha emissive feature was also observed in chiral aggregates with tubular shapes and was explained as follows[25,36]. The weak electrostatic interaction in the molecule aggregates results in an exciton energy splitting smaller than thermal energy at room temperature. Both excited states, i.e., $S_+$ and $S_-$, can therefore be significantly populated according to a Boltzmann probability distribution. The decay of $S_+$ and $S_-$ emits photons with opposite helicities and as a result yields the distinctive anti-Kasha bisignated CPPL signal. The same mechanism can explain the bisignate profiles of both CD and CPPL of the (432 helicoid IV NP)–(TDBC aggregate) hybrid films observed in our experiments. We thereby believe that the 432 helicoid IV NPs serve as the scaffolds that trigger the chiral arrangement of TDBC monomers and as a result promote the generation of chiral Frenkel excitons, although additional experimental evidence is needed to directly confirm the geometrical chiral arrangement of TDBC aggregates on the NP surface and to examine the energy splitting of formed chiral excitons in detail.

## Chiral-plasmon-dominated CPPL

In addition to serving as the scaffold that triggers the molecular chiral stacking and therefore promotes the generation of chiral excitons, chiral NPs have the potential to further improve the asymmetric CPL of their hybrids with molecule aggregates through optical chirality enhancement (see Supplementary Fig. 17 for the simulation results of 432-helicoid III). When the Au 432-helicoid IV NPs were replaced by the 432-helicoid III NPs with a strong far-field chiroptical response, the hybrids exhibited a completely different CPPL response. Their $g_{PL}$ spectra presented inverse spectral shapes compared to those of the extinction $g_{ext}$ spectra of the corresponding chiral plasmonic NPs (Fig. 3g). Same chiroptical response inversion was also observed in the CPPL of the hybrid films constructed from the 432 helicoid III NPs and various Rhodamine molecules. The 432 helicoid III NPs with $g_{ext} = −0.08$ at 590 nm resulted in the hybrid films showing fluorescent $g_{PL} = +0.017$ at 582 nm, +0.023 at 610 nm, +0.013 at 620 nm when the NPs were hybridized with R590, TDBC aggregates, and R640, respectively (Supplementary Fig. 18). Moreover, control experiments on the hybrid films of the silica-shell-coated 432 helicoid III NPs and TDBC aggregates showed rarely changed CPPL as the thickness of the silica shell was increased from 0 to 80 nm (Supplementary Fig. 19). The emission feature of chiral excitons was absent, and the CPPL was believed to mainly come from a plasmonic polarization-filtering effect, i.e., circular-polarization-preferential scattering and absorption of unpolarized photons emitted from achiral emission centers by the

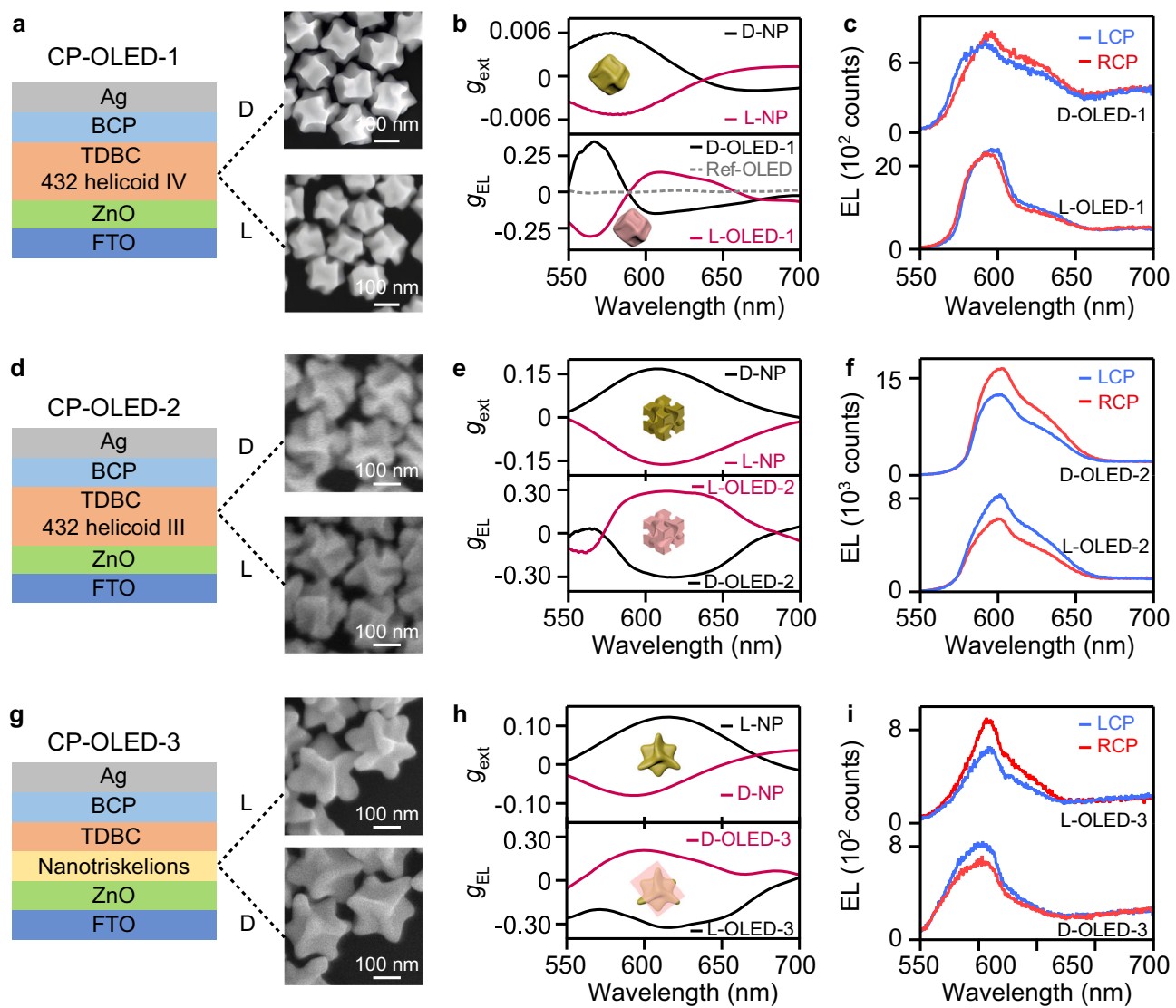

**Fig. 4 | Chiral electroluminescent emissions mediated by chiral Frenkel excitons and plasmonic polarization-filtering effect in the CP-OLEDs. a–c** CP-OLED-1 devices. The D- and L-432 helicoid IV NPs were utilized to prepare the (chiral plasmonic NP)–(TDBC aggregate) hybrid films working as the emissive layers to fabricate the D- and L-OLED-1 devices, respectively (**a**). The chiral emissions mediated by chiral Frenkel excitons are demonstrated by the weak CD response of the 432-helicoid IV NPs (**b**, top), the smoothed electroluminescent $g_{EL}$ spectra (**b**, bottom), and the CPEL spectra of the CP-OLED-1 devices (**c**) under the applied voltage of $U = 6$ V. **d–f** CP-OLED-2 devices. The D- and L-432 helicoid III NPs with strong far-field chiroptical responses were utilized to prepare the (chiral plasmonic NP)–(TDBC aggregate) hybrid films working as the emissive layers for the CP-OLED-2 devices, respectively (**d**). The chiral emissions originating mainly from the plasmonic polarization-filtering effect are demonstrated by the strong CD response of the 432-helicoid III NPs (**e**, top), the smoothed electroluminescent $g_{EL}$ spectra (**e**, bottom), and the CPEL spectra of the corresponding CP-OLED-2 devices (**f**) at an applied voltage $U = 9$ V. **g–i** CP-OLED-3 devices. The Au nanotriskelions were used as a separate layer in the CP-OLED-3 devices (**g**). The chiral emissions mediated only by the plasmonic polarization-filtering effect is demonstrated by the strong CD response of the nanotriskelions (**h**, top), the smoothed electroluminescent $g_{EL}$ spectra (**h**, bottom), and the CPEL spectra of the corresponding CP-OLED-3 devices (**i**) at an applied voltage $U = 5$ V. The originally measured electroluminescent $g_{EL}$ spectra were shown in Supplementary Fig. 24. Source data are provided as a Source Data file.

chiral plasmonic NPs, as reported in ref. 38. These CPPL measurement results support the perspective of employing the (432 helicoid III)–(TDBC aggregate) hybrids as the electroluminescent emissive layer of the CP-OLED based on chiral plasmons.

**Chiral-exciton-dominated CP-OLEDs**
We fabricated proof-of-concept CP-OLED devices using an inverted device structure composed of fluorine-doped tin oxide (FTO), zinc oxide (ZnO), the emissive layer, bathocuproine (BCP), and the Ag layer. (see Fig. 4, Supplementary Fig. 20, and "Methods")[39,40]. We measured the CPEL spectra, transient EL spectra, and EQEs of the devices and characterized their circular polarized emission capability with the $g_{EL}$

factor (Supplementary Figs. 21–23). $g_{EL}$ has the same definition as $g_{PL}$, i.e., $g_{EL} = 2(I^{E}_{LCP} - I^{E}_{RCP})/(I^{E}_{LCP} + I^{E}_{RCP})$, where $I^{E}_{LCP}$ and $I^{E}_{RCP}$ are the LCP and RCP EL intensities, respectively. The (432 helicoid IV NP)–(TDBC aggregate) hybrid film was deposited as the emissive layer in the prepared CP-OLED-1 device (Fig. 4a). The CP-OLED-1 devices exhibited bisignate $g_{EL}$ spectra because of the chiral-exciton-induced wavelength discrepancy between the spectral peaks of the LCP and RCP EL signals (Fig. 4c, Supplementary Fig. 24). When a voltage of $U = 6$ V was applied, the D-OLED-1 device with the D-432 helicoid IV NPs in the hybrid emissive layer showed a positive $g_{EL}$ peak of +0.33 at 570 nm and a negative $g_{EL}$ peak of −0.15 at 606 nm, respectively. The L-OLED-1 device using the L-432 helicoid IV NPs showed a mirror-symmetric $g_{EL}$

spectrum. As a control, we also prepared Ref-OLED devices employing the TDBC J-aggregate film alone as the emissive layer. The Ref-OLEDs exhibited a hardly observed chiral EL feature (Fig. 4b, Supplementary Fig. 21). Our results clearly demonstrate the efficient excitation of chiral excitons by charge injection in the CP-OLED-1 devices.

We believe the much larger value of electroluminescent $g_{EL}$ than that of fluorescent $g_{PL}$, when the same (432 helicoid IV NP)–(TDBC aggregate) hybrid films are employed as the emissive layer, should be attributed to the more efficient generation of chiral excitons through charge injection. When the hybrids are excited optically, the chromophores far away from the plasmonic NPs make the almost achiral contribution to the PL signal. In comparison, in the electrically excited hybrids, charges will preferentially inject into the molecule aggregates on the metal NP surface because of the high electron conductivity of Au, giving rise to a much more efficient generation of chiral excitons than achiral excitons. The molecules supporting achiral excitons far away from the metal NPs can be viewed as in parallel connection with the (metal NP–molecule) hybrids. Because of the high electron conductivity of the metal NPs, the current injected into the (metal NP–molecule) hybrids is much larger than that injected into the molecules supporting achiral excitons. Despite the chirality reversion of the emissions introduced by electrode reflection[8], the backward-reflected emissions are attenuated by the plasmonic NPs embedded in the emissive layer. The resultant overall $g_{EL}$ is therefore distinctly higher than $g_{PL}$. In addition, we have observed that $g_{EL}$ further increases with increasing applied voltage $U$ in the CP-OLED-1 device (Supplementary Fig. 25). Owing to the plasmonic Purcell effect, chiral excitons near the NP surface radiatively decay in a faster manner that that of achiral excitons far away from the NPs. As $U$ is enlarged to a critical value $U_{sa}$, the number of electrically excited achiral excitons reaches a saturated value, i.e., all achiral excitons are excited simultaneously in the radiative lifetime of achiral excitons. Further increasing $U$ will not increase the number of achiral excitons. In contrast, excited chiral excitons near the NP surface can still become enriched when $U$ is larger than $U_{sa}$, since chiral excitons near the metal surface decay faster. The superior enrichment of chiral excitons at high voltages gives rise to distinguished CPEL performances. We also performed the transient CPEL measurement and observed and a gradually enlarged splitting between the LCP and RCP EL peaks (Supplementary Fig. 25), indicating that the increase of injected electron–hole pairs can guide the generation of chiral excitons with increased numbers.

### Chiral-plasmon-dominated CP-OLEDs

We prepared OLED devices (CP-OLED-2, Fig. 4d) with the (Au 432 helicoid III NP)–(TDBC aggregate) hybrid film as the emissive layer, in which the NPs support strong chiroptical resonance[41]. The D-type device (D-OLED-2) employing the D-432 helicoid III NPs with $g_{ext} = 0.17$ at 610 nm exhibited a $g_{EL} = -0.31$ at $U = 9$ V (Fig. 4e, Supplementary Fig. 24). The L-432 helicoid III NPs and corresponding emitting device (L-OLED-2) exhibited mirror-symmetrical $g_{ext}$ and $g_{EL}$ spectra. Both L- and D-type CP-OLED-2 devices displayed distinctly different LCP and RCP EL intensities (Fig. 4f). The CPEL results demonstrate that the emissions from the CP-OLED-2 devices are dominated by the plasmonic polarization-filtering-induced asymmetric emissions and less caused by the asymmetric emissions of chiral excitons.

We believe that different chiral NPs behave differently in directing the chiral assembly of molecule aggregates on their surfaces, resulting in varying efficiencies in generating chiral excitons. In contrast to the Au 432 helicoid IV NPs used in CP-OLED-1, the Au 432 helicoid III NPs employed in CP-OLED-2, in spite of their stronger far-field chiroptical response, exhibit inferior performance in producing chiral excitons in the molecule aggregates on their surfaces. The luminescence mediated by plasmon–exciton coupling in the (chiral plasmonic NP)–(TDBC aggregate) hybrids in CP-OLED-2 is therefore overwhelmed by the emissions modified by the plasmonic polarization-filtering effect.

Further study on individual hybrid nanostructures that can exclude the polarization-filtering effect is required to reveal how the (chiral plasmon)–(chiral exciton) coupling modulates the chiral emissions.

The emission dissymmetry factor $|g_{PL}|$ or $|g_{EL}|$ of the prepared (chiral plasmonic NP)–(TDBC aggregate) hybrids in CP-OLED-2 is much higher than those of most induced CPL systems[42]. Given that the chiral NPs prefer not to orient unidirectionally because of the stochastic movement of the NPs during the emissive layer preparation, the plasmonic polarization-filtering effect can further be improved by aligning the NPs to their axis showing the largest chiroptical response. We therefore constructed a third type of CP-OLED device (CP-OLED-3) with the emissive layer obtained by assembling a chromophore film and a layer of aligned chiral NPs that were prepared separately (Fig. 4g). The plate-like chiral Au nanotriskelions that can easily be laid on flat surfaces in an up-and-down direction were chosen and deposited to form large-area and high-density random NP arrays[16]. The CP-OLED-3 devices, in which only the plasmonic polarization-filtering effect contributes to the device CPEL owing to the rare existence of chiral excitons, exhibited similar CPEL performance as that of CP-OLED-2 (Fig. 4h, i). The L-type device (L-OLED-3) employing the L-nanotriskelions with $g_{ext} = 0.12$ at 610 nm exhibited a $g_{EL} = -0.32$ at $U = 5$ V. The chiral NP alignment did improve the OLED CPEL performance, as demonstrated by the much larger $|g_{EL}|$ and $|g_{PL}|$ of the emissive layer than the $|g_{ext}|$ of the nanotriskelions in solution (Fig. 4h).

In all CP-OLED-2 and CP-OLED-3 devices, the EL emitted forward to the transparent fluorine-doped tin oxide (FTO) electrode or emitted backward and got reflected by the Ag back electrode[8]. We demonstrated that both the forward and backward-reflected emissions were circularly polarized by the chiral NPs to carry the same handedness, preventing any circular polarization cancellation and significantly boosting the asymmetric EL (Supplementary Figs. 26, 27)[43]. We have also fabricated the CP-OLED-2 and CP-OLED-3 devices using the chiral NPs with different $g_{ext}$-factors (Supplementary Figs. 28–33). The resultant CPEL dissymmetry of the CP-OLED-2 and CP-OLED-3 devices increased almost linearly with $|g_{ext}|$ of the corresponding chiral NPs, reaching values higher than those of CP-OLED-1 with strong asymmetric emissions from chiral excitons (Fig. 5a).

### Luminous efficiency of the CP-OLEDs

We additionally explored the energy conversion efficiency of our CP-OLEDs (Supplementary Fig. 23). The varying EQE–$U$ dependence (Fig. 5b) results from the different device configurations. The CP-OLED-1 and CP-OLED-2 devices shared the same device configuration and exhibited similar EQE–$U$ dependence. The EQEs of CP-OLED-1 and CP-OLED-2 presented a first increasing then decreasing feature as $U$ was increased. CP-OLED-3, using a separate nanotriskelion layer, exhibited a similar EQE–$U$ dependence to that of Ref-OLED and displayed monotonically decreasing EQEs as $U$ was increased. The CP-OLED-2 devices performed the best in the luminous efficiency, with the EQEs reaching 2.5% (Fig. 5b). The superior EQE performance of the CP-OLED-2 devices can be attributed to the following two reasons. The first is the large plasmonic near-field enhancement near the surface of the 432 helicoid III NP (Supplementary Fig. 34). The plasmonic near-field enhancement can accelerate the radiation process of excitons because of the Purcell effect[34], giving rise to a higher emission quantum efficiency. The plasmonic NPs further work as antennas that can increase the external extraction efficiency[44,45]. The second reason is the suppression of the overshoot effect that was observed in the Ref-OLED and CP-OLED-3 devices. The overshoot effect came from the radiative recombination of pre-stored electrons and injected holes, characterized by a spike with a super-high intensity at the transient EL rising edge (Supplementary Fig. 22)[46,47]. We found that the overshoot effect can make the OLEDs work in an unstable state, reducing the EQE and luminous lifespan (Supplementary Figs. 22b, 23). The transient EL examination of the CP-OLED-2 devices have revealed that the chiral

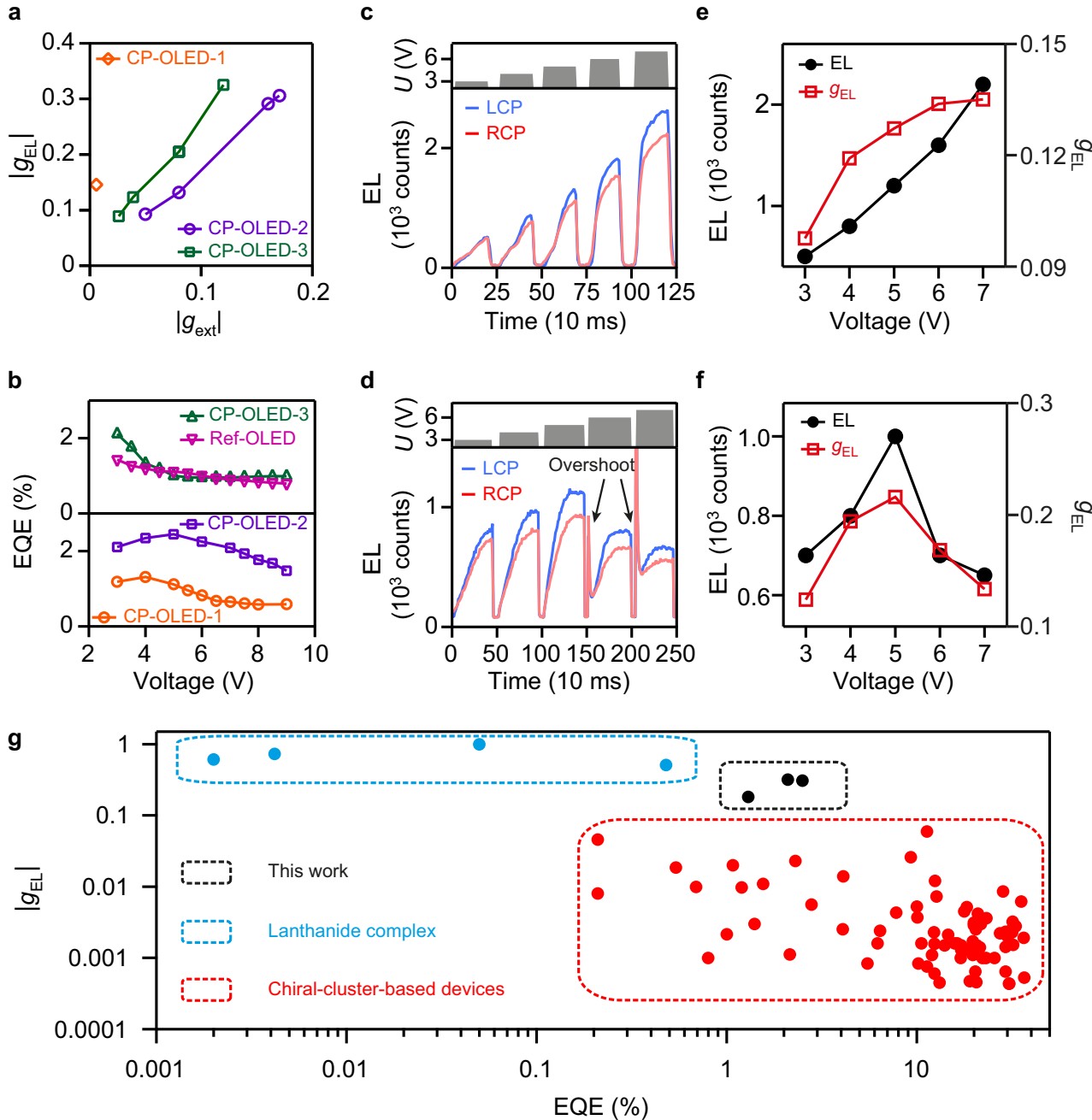

**Fig. 5 | Circular polarization dissymmetry and energy efficiency performance of the CP-OLEDs with different configurations. a** Dependence of the highest device |$g_{EL}$| of the CP-OLEDs on the |$g_{ext}$| of the employed chiral NPs. The utilization of the chiral NPs with increased |$g_{ext}$| factors contributes to the higher asymmetric EL emissions of the CP-OLED-2 and CP-OLED-3 devices, indicating that the CPEL asymmetry is governed by the chiroptical activity of the chiral NPs. **b** EQEs of the CP-OLED devices at different applied voltages. **c–f** Transient CPEL intensities (exposure time 10 ms) of typical L-OLED-2 (**c**) and D-OLED-3 (**d**) when the applied voltages were swept between zero and linearly increasing values. The dependences of the EL intensity and $g_{EL}$ on the applied voltage for the same L-OLED-2 (**e**) and D-OLED-3 (**f**) devices were derived accordingly. **g** Comparison of various CP-OLEDs in terms of EQE value and |$g_{EL}$| factor. The data of previously reported CP-OLEDs were summarized in Supplementary Table 1. Different from previously reported CP-OLEDs showing the inverse relationship between the EQE value and |$g_{EL}$| factor, the CP-OLEDs constructed from the chiral plasmonic NP−molecule hybrids exhibit both high EQE values and large |$g_{EL}$| factors. Source data are provided as a Source Data file.

NPs with good electrical conductivity is helpful to significantly reduce the amount of pre-stored electrons, eliminating the overshoot effect (Supplementary Fig. 22).

In addition, as $U$ became larger, the CP-OLED-2 devices exhibited gradually increasing |$g_{EL}$| that tended to reach saturation (Fig. 5c,e, Supplementary Figs. 28–30), while |$g_{EL}$| of CP-OLED-3 reached its maximum value before suffering from an overshoot-effect-induced degradation (Fig. 5d,f, Supplementary Figs. 31–33). The different |$g_{EL}$|−$U$ evolution behaviors imply that in addition to far-field plasmonic polarization-filtering, the plasmonic near-field-modulated emissions of chiral excitons also contribute to the CPEL of the CP-OLED-2 devices. Plasmonic polarization-filtering achieves overwhelming dominance in CP-OLED-2 at high voltages. These results indicate that the involvement of chiral NPs can significantly improve

the chiroptical and electrical performance in the CPEL process and thereby overcome the current challenge of CP-OLEDs (Fig. 5g). Previously reported CP-OLEDs generally present the inverse relationship between the EQE value and $|g_{EL}|$ factor (Supplementary Table 1)[7,48,49]. For example, lanthanide complex-based CP-OLEDs show extremely high $|g_{EL}|$ factors but low EQE values, while chiral cluster-based CP-OLEDs generally exhibit high EQE values and low $|g_{EL}|$ factors. In contrast, the CP-OLEDs in our work can achieve both high EQE and $|g_{EL}|$ values. The EQEs of our devices are about two orders of magnitude higher than those of lanthanide-complex-based devices and the $|g_{EL}|$ factors are about two orders of magnitude larger than those of chiral-cluster-based devices.

## Discussion

To summarize, we have developed circularly polarized OLEDs exhibiting extraordinary CPEL asymmetry ($g_{EL} \approx 0.31$) and luminous efficiency (EQE $\approx 2.5\%$) based on (chiral plasmonic NP)–(molecule aggregate) hybrids. The chiral NPs serve as scaffolds directing the chiral stacking of molecule aggregates to generate chiral excitons and plasmonic nanoantennas polarizing emitted photons. Designing (chiral NP)–(molecule aggregate) hybrids with both strong chiral excitonic and plasmonic properties will bring further improvement of the circular polarization asymmetry in the EL emissions[50]. We are confident that with the help of more advanced preparation strategy of the emissive layer based on NP–molecule hybrids, as well as the fast-developing techniques of growing chiral plasmonic NPs, the employment of (chiral plasmonic NP)–molecule hybrids will lead to CP-OLEDs with outstanding circular polarization emission dissymmetry and efficiencies. The proposed method only involves the design of the emissive layer and does not require any specialized changes on the OLED device architecture or molecular modification on the chromophore. Our strategy is therefore compatible with the current manufacturing technology of OLEDs and can also be applied to devices with different structures[39]. In addition, the capability of chiral plasmonic NPs in controlling the chiroptical properties of supramolecular chromophores enables rich chiral plasmon–exciton interactions at the single-nanostructure level, making chiral plasmonic NPs promising candidates for constructing other chiral photoelectric devices, such as non-reciprocal nanoscale light sources and deterministic spin–photon interfaces for quantum networks[51,52], without the need for precision design and complex manufacturing.

## Methods

### Materials

Ascorbic acid (AA, 99%), L-glutathione (L-GSH, 98%), ammonia solution (NH$_3 \cdot$H$_2$O, in water, 35 wt.%), polyvinyl alcohol (PVA, $M_w$: 146–186 kg mol$^{-1}$, 87–89%, hydrolyzed), thiol-modified poly(ethylene glycol) (mPEG-SH, average $M_n$: 2 kg mol$^{-1}$), zinc acetate (Zn(CH$_3$COO)$_2 \cdot$2H$_2$O), ethanolamine (99%), and 2-methoxyethanol (99.8%) were purchased from Sigma-Aldrich. Gold(III) chloride trihydrate (HAuCl$_4 \cdot$3H$_2$O, 99.9%) were purchased from Tianjin Jinbolan Fine Chemical Co. Ltd. D-glutathione (D-GSH, 99%) was purchased from Apeptide Co. Ltd. (Shanghai, China). Cetyltrimethylammonium bromide (CTAB, 98%) was obtained from Alfa Aesar (Batch number: A15235). Potassium iodide (KI, 99%) and bathocuproine (BCP, 98%) were purchased from Aladdin Reagent Co. Ltd. (Shanghai, China). Tetraethyl orthosilicate (TEOS, 98%) was purchased from Acros Organics. The chromophore, 5,6-dichloro-2-[[5,6-dichloro-1-ethyl-3-(4-sulfobutyl)-benzimidazol-2-ylidene]-propenyl]-1-ethyl-3-(4-sulfobutyl)-benzimidazolium hydroxide (TDBC, inner salt) was purchased from Shanghai Ruipu Chemical Technology Co. Ltd. Rhodamine 640 perchlorate (R640) and Rhodamine 590 perchlorate (R590) were purchased from Exciton-Luxottica Inc. Deionized (DI) water with a resistivity of 18.2 M$\Omega$ cm was produced by a Direct-Q 5 ultraviolet water purification system.

### Synthesis of the chiral nanoparticles

A GSH-directed and halide-assisted differential growth strategy[16] was employed to guide the chiral growth on Au nanocubes, nanooctahedrons, and nanodisks into 432 helicoid IV, 432 helicoid III, and nanotriskelions, respectively (Supplementary Figs. 1, 2). The nanocrystal seeds (nanocubes, nanooctahedrons, and nanodisks) were synthesized by seed-mediated growth. The optical densities of the as-prepared nanocrystal seed solutions at their plasmon peaks were adjusted to be 0.8 (optical path length: 1.0 cm) by adding DI water before use. The growth solution was prepared by first adding CTAB (0.1 M), KI (0.01 M), GSH (2.75 mM), and HAuCl$_4$ (0.01 M) solutions of varying volumes into DI water sequentially. The mixed solution turned colorless after the subsequent addition of AA (0.1 M). The seed solution (2 mL) was rapidly injected into the colorless mixture under stirring. The resultant mixture (total volume: 10 mL) was kept in an isothermal oven at 35 °C for 2 h. The obtained chiral nanoparticles (NPs) were washed by centrifugation and redispersed in CTAB solutions (1 mM, 1 mL). The chiral NP solutions were stored in the environment of 0–5 °C. The amounts of CTAB and KI in the growth solution were varied to control the morphologies of the chiral NPs. A library of chiral NPs with helicoidal geometry and controllable sizes were as a result prepared. These NPs exhibited chiral plasmonic properties with different CD responses. For example, L-432 helicoid III and L-nanotriskelions, both of which were obtained when L-GSH was employed as the chiral ligand during synthesis, exhibited opposite geometric chirality and chiroptical responses that resulted from the halide-assisted favored growth along different crystal axes.

### Coating of silica shell on the 432 helicoid III NPs

The coating of silica shell was achieved through the hydrolysis and condensation of the silica precursor, TEOS, on the surface of the synthesized NPs. Specifically, the synthesized 432 helicoid III NPs were first dispersed in DI water and mixed with mPEG-SH (1 mM, 1 mL) under stirring for 24 h. The NPs coated with mPEG-SH were washed and redispersed in the silica precursor solution, which was prepared in advance by mixing ethanol (3.75 mL), DI water (1.125 mL), NH$_3$·H$_2$O (35 wt.%, 75 μL), and TEOS (10 vol.%, in ethanol, 20 μL). The mixture was subjected to ultrasonic treatment in an ice water bath for 90 min for silica coating. The resultant mixture solution was centrifuged, followed by redispersion of the precipitate in ethanol. In this way, silica shell can be coated uniformly onto the surface of each Au NP. A larger amount of TEOS in the silica precursor solution gave rise to a thicker silica shell (Supplementary Fig. 19).

### Deposition of aligned Au nanotriskelions densely on substrates

The synthesized nanotriskelions were washed and dispersed in an aqueous CTAB solution with its concentration lower than 5 μM. Glass slides were used as substrates for the deposition of the Au nanotriskelions. They were cleaned in ethanol under ultrasonic treatment for 30 min and treated with oxygen plasma for 3 min. The cleaned glass slide was immersed into the Au nanotriskelion solution for 24 h. The Au nanotriskelions were therefore randomly deposited onto the glass slide through an electrostatic interaction, resulting in a large-area and high-density random NP array on the substrate (Supplementary Fig. 27).

### Preparation of the (plasmonic NP)–molecule hybrid films

To prepare the (chiral plasmonic NP)–(TDBC aggregate) hybrids, the mixture solution of the colloidal chiral NPs and TDBC monomers were dropped onto transparent glass or conductive substrates placed on a hot plate set at 45 °C (Supplementary Fig. 5). The evaporation of water drove the assembly of TDBC on the surface of the chiral NPs, forming (chiral plasmonic NP)–(TDBC aggregate) hybrid films. In total, 50 μL of the chiral NPs and TDBC with a specific volume (0.92 mM) were used. The structural chirality of the chiral NPs was transferred to the TDBC

aggregates coated on the NPs. The control samples of (Au nanosphere)–(TDBC aggregate) hybrids were prepared in the same way by replacing the chiral Au NPs with achiral Au nanospheres. In contrast, (plasmonic chiral NP)–Rhodamine hybrids were prepared by embedding the chiral NPs into the PVA matrix containing R590 and R640. Specifically, PVA powder (1.3 g) was dissolved in DI water (40 mL) and subjected to 90 °C oil-bath heating under stirring. The PVA solution mixed with the colloidal NP and Rhodamine solutions was then dropped onto the substrate placed on the hot plate set at 45 °C. After water evaporation, the (plasmonic chiral NP)–Rhodamine hybrid film was thus obtained.

## OLED fabrication
The proof-of-concept OLED devices were fabricated using an inverted device structure composed of FTO, ZnO, the emissive layer, BCP, and a silver layer. ZnO and BCP were selected as the electron and hole transport layers, respectively[39,40]. The preparation process of the OLED devices followed the steps shown in Supplementary Fig. 20a. The FTO strip electrode of 2 mm in width was washed by sequential ultrasonication in detergent, deionized water, acetone, and isopropanol for 20 min each, and dried on a hot plate before being cleaned by ultraviolet ozone for 20 min. After being dried, the FTO electrode was spin-coated with the ZnO precursor solution at 2000 rpm for 30 s and then annealed at 200 °C for 30 min to obtain the electron transport layer. The ZnO precursor solution was prepared by adding $Zn(CH_3COO)_2 \cdot 2H_2O$ (100 mg) and ethanolamine (28.29 μL) into 2-methoxyethanol (973 μL) and stirred overnight before use. The ZnO layer was then covered by adhesive tapes, to form a channel of 12 mm in length and 4 mm in width on top of the FTO electrode. To prepare the Ref-OLED device, TDBC aqueous solution (0.92 mM, 80 μL) was drop-cast onto the ZnO layer. The substrate was spun at a low speed of 300 rpm and sequentially placed onto a hot plate at 45 °C. The TDBC aggregate film with high uniformity was produced after water evaporation, forming an emissive layer with a thickness of 560 ± 37 nm. To prepare the emissive layers of the CP-OLED-1 and CP-OLED-2 devices, the TDBC aqueous solution (0.92 mM, 80 μL) was first mixed with the 432 helicoid IV NP solution (100 μL) and the 432 helicoid III NP solution (50 μL) before the drop-casting process, respectively. To prepare the emissive layer of the CP-OLED-3 device, the Au nanotriskelions were first deposited on the FTO/ZnO electrode by drop-casting to form a layer of a high-density random NP array. The TDBC aggregate film was then prepared separately on top of the NPs following the same procedure of preparing the emissive layer of Ref-OLED. On top of the prepared emissive layer, the hole transport layer was deposited by drop-casting the BCP solution under rotation at a speed of 500 rpm and a sequential drying process. The BCP solution was prepared by dissolving BCP (0.1 g) in isopropyl alcohol (40 mL) under stirring for 24 h. The devices were then stored in a glove box with nitrogen environment and then evaporated with Ag electrodes of 2 mm in width through a shadow mask.

## Instrumentation
SEM images were taken on a JEOL JSM 7800 F microscope at an operation voltage of 10 kV. Extinction spectra were measured on an ultraviolet/visible/near-infrared spectrophotometer (PerkinElmer Lambda 950). Extinction CD spectra were recorded on a CD spectrophotometer (JASCO J-1500). PL and CPPL spectra were acquired on a CPL spectrometer (JASCO CPL-300). ORD spectra were calculated from the CD spectra with Kramers-Kronig transformation, as described by ref. 33

$$\varphi(\lambda) = \frac{2}{\pi} \int_0^\infty \theta(\mu) \frac{\mu}{\lambda^2 - \mu^2} \, d\mu \qquad (1)$$

where $\varphi(\lambda)$ is the molar rotation (as a function of wavelength $\lambda$) and $\theta(\mu)$ is the molar ellipticity (as a function of wavelength $\mu$).

## CPEL measurement
CPEL spectroscopy was performed on a home-built optical setup. As shown in Supplementary Fig. 21, two sets of circular polarization plates were employed, each of which was composed of a quarter-wave plate (Union Optic Inc., 550–750 nm) and an ultrathin linear polarizer (Union Optic Inc., 550–900 nm). These two plate groups were integrated into one bracket, one for measuring the LCP component and the other for recording the RCP component of the emission. The intensity of the emitted EL from the OLED device passing through the plate groups was measured by the spectrometer system composed of a charge coupled device (CCD) camera (Princeton Instruments, Pixis 400) and a monochromator (Acton, SpectraPro 2360i). The OLED devices were connected with a Source Meter instrument (Keithley 2636B), which provided voltage pulses. Voltage pulses with a 2-s interval were applied to accurately measure the intensity of CPEL components. During a 2-s interval, two plate groups were rapidly moved to switch the LCP and RCP component measurement. We can therefore record the LCP and RCP components under different voltage pulses. Various OLED devices of the same type were examined to ensure the accuracy of the CPEL measurements (Supplementary Fig. 21). We first prepared Ref-OLED devices employing the TDBC J-aggregate film alone as the emissive layer. The TDBC J-aggregate film showed a weak chiroptical response and the Ref-OLEDs exhibited a nearly achiral EL feature. Under two cycles of CPEL measurement, we recorded the transient profiles of the EL intensity, CPEL spectra, and $g_{EL}$ spectra, demonstrating the unpolarized EL emitted from the Ref-OLED device. We also measured the CPEL profiles from our CP-OLED devices. For example, the CP-OLED-3 devices displayed distinctly different LCP and RCP EL intensities.

We would point out that the fluorescence CPL and extinction CD measurements were performed on commercial CPL and CD spectrometers that collect differential circular polarization signals in a wavelength-by-wavelength manner by switching the circular polarization between RCP and LCP accurately for each individual scanning wavelength. In contrast, the CPEL characterization was carried out in our home-built setup that employs a broadband linear polarizer and quarter-wave plate. The difficulty in controlling polarization perfectly at a broad wavelength range in the CPEL measurements leads to errors in the obtained $g_{EL}$ spectra. The largest error can be estimated to be $|\Delta g_{EL}| \approx 5\%$ at $|g_{EL}| \approx 30\%$ when the phase retardance $\phi$ between two orthogonal linear eigen-polarizations introduced by the quarter-wave plate has a fluctuation of ±10% around the expected value π/4. The EQE measurements were carried out by collecting the emitted EL light with an optical fiber (QS-600-1.5PVC, Ocean Optics) and directing the EL to a spectrometer (QE65PRO, Ocean Optics). The collected photon counts $N$ were therefore recorded. The current ($J$) was simultaneously measured by an ammeter when different voltages were applied to the OLED device. The EQE was therefore calculated according to EQE = $(N_{ph} e R)/J$, where $e$ is the elementary charge, $R$ is the ratio of the OLED device area to the cross-section of the fiber core, and $N_{ph}$ is the total number of photons emitted by the OLED and collected by the optical fiber per second. $N_{ph} = N/\eta$, where $\eta$ is the detection efficiency defined as the ratio of the number of detected photons to the number of emitted photons. In our setup, $\eta \approx 43\%$ considering the collection loss.

## Electrodynamic simulations
Finite-difference time-domain (FDTD) simulations were performed using FDTD Solutions (Lumerical 8.19) to calculate the optical responses of the chiral plasmonic NPs. Circularly polarized light was modeled by the superposition of two orthogonally polarized plane light waves with the same position and a phase difference of ±π/2. LCP and RCP light were launched onto a single chiral NP to simulate the

interaction. The wavelength range of the light source was set at 400–800 nm. A mesh size of 1.0 nm was used. The refractive index of the surrounding was set at 1.0 for air. The dielectric function of Au was obtained from the measured data of Johnson and Christy. To verify the chiroptical response of the chiral NP, three computational quantities, the dissymmetry factors of the NP absorption, scattering, and extinction ($g_{\text{cal-abs}}$, $g_{\text{cal-sca}}$, $g_{\text{cal-ext}}$) were calculated, as described by $g_{\text{cal}} = 2(\sigma_{\text{LCP}} - \sigma_{\text{RCP}})/(\sigma_{\text{LCP}} + \sigma_{\text{RCP}})$. $\sigma_{\text{LCP}}$ and $\sigma_{\text{RCP}}$ represent the calculated absorption/scattering/extinction cross-sections under the illumination of LCP and RCP light, respectively. The spectra of the absorption, scattering, and extinction $g_{\text{cal}}$ for the different chiral NPs were calculated, as shown in Supplementary Fig. 3. The optical chirality $C(\mathbf{r})$ describes the electromagnetic density of chirality. It can be derived from the calculated electric field $\mathbf{E}(\mathbf{r})$ and magnetic field $\mathbf{B}(\mathbf{r})$ according to $C(\mathbf{r}) = -\varepsilon_0\omega\text{Im}[\mathbf{E}^*(\mathbf{r})\cdot\mathbf{B}(\mathbf{r})]/2$, where $\varepsilon_0$ is the permittivity of vacuum and $\omega$ is the frequency of the incident light. The optical chirality enhancement is thus written as $C(\mathbf{r})/|C_{\text{CPL}}(\mathbf{r})| = -c_0\text{Im}[\mathbf{E}^*(\mathbf{r})\cdot\mathbf{B}(\mathbf{r})]/|\mathbf{E}_{\text{CPL}}(\mathbf{r})|^2$, where $C_{\text{CPL}}$ is the value obtained for circularly polarized light without the nanostructure, $\mathbf{E}_{\text{CPL}}$ is the electric field of circularly polarized light, and $c_0$ is the speed of light. $C(\mathbf{r})/|C_{\text{CPL}}(\mathbf{r})|$ is $-1$ for LCP light and $1$ for RCP light. Superchiral near-field can be demonstrated when $|C(\mathbf{r})|>|C_{\text{CPL}}(\mathbf{r})|$. We calculated the distributions of the optical chirality enhancement for the various chiral NPs under the excitation of LCP and RCP light, as shown in Supplementary Fig. 17. We also calculated the absorption/scattering cross-sections and distributions of plasmonic enhancement $|E|^2/|E_0|^2$ at the surface of Au 432 helicoid III, as shown in Supplementary Fig. 34.

## Theoretical model

We developed a theoretical model to evaluate the optical responses of the chiral plasmonic–excitonic systems (Supplementary Fig. 8). Similar to chiral molecules, the chiroptical property of a chiral plasmonic NP can be described by dipole approximation. The effective electric dipole $\mathbf{p}_1$ and effective magnetic dipole $\mathbf{m}_1$ of a small NP can be written as

$$\mathbf{p}_1 = \alpha\mathbf{E}_1 - iG\mathbf{H}_1; \mathbf{m}_1 = \chi\mathbf{H}_1 - i\bar{G}\mathbf{E}_1 \tag{2}$$

where $\mathbf{E}_1$ and $\mathbf{H}_1$ are the local electric and magnetic fields at the chiral NP, respectively. The effective polarizability $\alpha$ (to the first order of the chiral parameters $c$ and $\bar{c}$) can be calculated by

$$\alpha = \frac{V}{L + \left(\frac{\varepsilon_0}{\varepsilon_{\text{m}} - \varepsilon_0}\right) - \frac{i2k^3V}{3}} \tag{3}$$

where $\varepsilon_0$ and $\varepsilon_{\text{m}}$ are the dielectric constants of the background and the metal, $V$ is the volume of the NP, and $k$ is the wavevector. The mixed electric-magnetic polarizabilities $G = c\alpha/(\varepsilon_{\text{m}} - \varepsilon_0)$ and $\bar{G} = \bar{c}\alpha/(\varepsilon_{\text{m}} - \varepsilon_0)$. $\chi = \tilde{c}V$. For a NP that can be modeled as a prolate spheroid with semi-major axis $a$ and semi-minor axis $b$, the geometric factor $L = (1/e^2-1)\{1/(2e)\cdot\ln[(1+e)/(1-e)]-1\}$, where the eccentricity $e = [1 - (b/a)^2]^{1/2}$. For a NP that can be modeled as a sphere, $L = 1/3$. Similarly, the optical response of a chiral molecular J-aggregate is determined by its electric and magnetic dipoles, which can be calculated based on the following equations

$$\mathbf{p}_2 = \varepsilon_{\text{b}}\mathbf{E}_2 - i\kappa_{\text{b}}\mathbf{H}_2; \mathbf{m}_2 = \mu_{\text{b}}\mathbf{H}_2 - i\bar{\kappa}_{\text{b}}\mathbf{E}_2 \tag{4}$$

where $\mathbf{E}_2$ and $\mathbf{H}_2$ are the local electric and magnetic fields at the molecular J-aggregate. For chiral molecules, $\mu_{\text{b}}$ is usually very small ($\mu_{\text{b}} \approx 0$), $\varepsilon_{\text{b}} = \varepsilon_{\text{b0}} - c_2\cdot f$, $\kappa_{\text{b}} = \kappa\cdot f$, $\bar{\kappa}_{\text{b}} = \bar{\kappa}\cdot f$, $f = \omega_0^2/(\omega^2 - \omega_0^2 + i\omega\gamma_0)$. In the presence of interaction between the NP and molecules, the local electric field or magnetic field of the NP is the superposition of the incident field and that generated by the electric or magnetic dipole of the molecules. The local electric field or magnetic field of the

molecules is the superposition of the incident field and that generated by the electric or magnetic dipole of the NP.

The CD spectrum of the NP–molecule hybrid can be calculated by the formula

$$\text{CD} = \text{Im}[\mathbf{E}_0^* \cdot (\mathbf{p}_1 + \mathbf{p}_2) + \mathbf{H}_0^* \cdot (\mathbf{m}_1 + \mathbf{m}_2)] \tag{5}$$

The Drude model was used to describe the metal permittivity $\varepsilon_{\text{m}} = \varepsilon_\infty - \omega_{\text{p0}}^2/(\omega^2 + i\omega\gamma_{\text{p0}})$, where $\omega_{\text{p0}}$ and $\gamma_{\text{p0}}$ are the volume plasmon frequency and relaxation constant of the metal. We can thus obtain $(\omega - \omega_0)(\omega - \omega_{\text{p}}) - g^2 = 0$, with the plasmon resonance frequency of the NP $\omega_{\text{p}} = \omega_{\text{p0}}/[\varepsilon_\infty + \varepsilon_0\cdot(1 - L)/L]^{1/2}$.

## Reporting summary

Further information on research design is available in the Nature Portfolio Reporting Summary linked to this article.

## Data availability

The data that support the findings of this study are available from the corresponding authors upon request. Source data are provided with this paper.

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

## Acknowledgements

This project received support from the National Natural Science Foundation of China (12088101 (H.-Q.L.), 62375290 (L.S.), 12174032 (W.Z.). J.P.Z. acknowledges support from the Shanghai Municipal Science and Technology Major Project. L.S. acknowledges support from the Pearl River Talent Recruitment Program (2019QN01C216) and the Science and Technology Planning Project of Guangdong Province (2023B1212060025). J.F.W. acknowledges support from the Research Grants Council of Hong Kong (ANR/RGC, A-CUHK404/21).

## Author contributions

J.P.Z., L.S., and J.F.W. conceived the project. J.P.Z. performed the NP synthesis, CD, CPPL, and CPEL measurements and electron microscopy characterization. J.P.Z., Y.A.F., and X.H.L. fabricated the OLED devices. J.W., W.Z., and H.-Q.L. performed electromagnetic modeling. J.P.Z., L.S., and J.F.W. wrote the manuscript. All authors commented on the manuscript. J.F.W. and L.S. supervised the project.

## Competing interests

The authors declare no competing interests.

 
