## [Transparent Peer Review file · Nature Communications]

Circularly polarized OLEDs from chiral plasmonic nanoparticle-molecule hybrids

Corresponding Author: Professor Jianfang Wang

Version 0:

Reviewer comments:

Reviewer #1

(Remarks to the Author)

Wang et al. report on the assembly of chiral plasmonic nanoparticles (NPs) and supramolecular aggregates to construct various chiral polymer organic light-emitting diodes (CP-OLEDs). The multiscale chirality transfer, plasmonic enhancement, and suppression of the overshoot effect in the aggregates containing chiral plasmonic nanoparticles and chromophores within the emissive layer yield an external quantum efficiency (EQE) of 2.4% and a gEL factor of 0.31. The work demonstrates that chiral plasmonic nanoparticles (NPs) can serve as promising candidates for ultracompact chiral photoelectric devices, presenting a novel concept worth publishing. However, several issues have been identified in the manuscript, as outlined below. I recommend that the paper be reevaluated once these problems have been addressed.

1. The authors mention the multiscale chirality transfer from chiral plasmonic nanoparticles to chiral plasmonic nanoparticles, but what is the medium of chiral transmission? The authors should be more deeply discussed.
2. Please provide a detailed explanation about the overshoot effect. Does it effect the gEL factor of OLED device?
3. The device preparation process should be described in detail. For example, how the emissive layer is prepared, what solvent is used, and what is the concentration of spirochaeta. How is the uniformity of the emissive layer directly dropped on the ZnO surface and whether it affects the effect of the OLED device?
4. More importantly, why the authors use this device structure rather than the commonly used ITO/PEDOT:PSS/HTL/EL/ETL/LiF/Al structure type? How did the author consider it?
5. Although the authors give a corresponding explanation for gEL larger than gPL, this is still unreasonable for the structure of electroluminescent devices. Whether it is chiral organics, complexes, or nanoclusters that are used as emissive layers, the gEL of the device is usually smaller than the gPL, the accepted reason is that the chirality will be reversed during the reflection process so that the gEL signals counteract each other. Why is it?
6. In the manuscript, the authors demonstrate that the CP-OLEDs exhibit efficient CP emission performance, achieving a high external quantum efficiency (EQE) of 2.4% and a considerable gEL factor of 0.31, as compared to the limited selection of CPEL devices presented in Supplementary Table 1. However, the authors have only compared a small number of CPEL devices, and the performance of the CP-OLEDs in this study does not appear particularly exceptional. Therefore, a more thorough discussion of the results is warranted.
7. I observe that the chiral plasmonic NPs possess > 100 nm size, and what is the thickness of the CP-OLED emissive layer? Will such an emission layer be detrimental to device performance?
8. The EQE of 2.4% for the OLED device is not competitive for measuring CPEL. As is widely recognized, testing CPEL presents significant challenges; therefore, the authors should provide detailed methodologies to ensure the accuracy of the test results.

Reviewer #2

(Remarks to the Author)

Reviewer's Report for:

"Circularly polarized OLEDs constructed from chiral plasmonic nanoparticle–molecule hybrids"

By: Jiapeng Zheng, Yuang Fu, Jing Wang, Wei Zhang, Xinhui Lu, Hai-Qing Lin, Lei Shao & Jianfang Wang

This is a nice report of chiroptical and electro-chiroptical properties of hybrid systems consisting chiral Au nanoparticles and of 5,6-dichloro-2-[[5,6-dichloro-1-ethyl-3-(4-sulfobutyl)-benzimidazol-2-ylidene]-propenyl]-1-ethyl-3-(4-sulfobutyl)-benzimidazolium hydroxide (TDBC) chromophore molecule. The manuscript reports on an exquisitely novel research possibly impacting on CD spectroscopy, sensing and photoluminescence. It deals with the interaction of plasmonic CD/CPPL/CPEL from Au NPs and of excitonic CD/CPPL/CPEL from TDBC. My approval to publication is quite likely, subject to the Authors' responses to the following questions/comments:

- 1) Is TDBC a non-chiral molecule? If so, is excitonic CD coming just from chiral arrangements around the chiral NPs? Is it the experience of this Reviewer that chirality of the organic molecule is a bonus for the production of CD spectra.
- 2) It is the experience of this Reviewer to define the circularly polarized emission as CPL (=circularly polarized luminescence) rather than CPPL (=circularly polarized photo-luminescence). Why did the Authors find it necessary to add an additional P to the mostly used CPL effect? Is CPPL an unknown effect to this Reviewer?
- 3) If CPPL is the same as CPL, due to Kasha's rule, then only the longest wavelength component of the "excitonic" CPL is active, as in part one may see from Figure 13 and following in Supplementary Material. (See the numerous review articles, part of which are even cited in this manuscript.)
- 4) A better description of the experimental chiroptical experiments should be provided in the Supplementary file (which apparatuses were employed, the spectroscopic resolution, the number of scans, etc.)
- 5) Minor points: (1) in the text please write, in more formal way, "TDBC concentration" rather than in the more concise way, [TDBC]. (2) The sentence on page 4 in the text "The observed bisignate CD spectrum, known as the Cotton effect,..." is inappropriate, since Cotton effect is an old way of calling CD, either monosignate or bisignate: Aimé Auguste Cotton lived in the eighteenth/nineteen centuries, much before any CD experimentation, when only the optical rotation was known and CD was deduced therefrom. (3) On Page 7: what do the Authors mean by "more chiral excitons"? (4) On page 11: what do the Authors mean by "CP degrees"? Do they mean: "CP intensity"?

Reviewer #3

(Remarks to the Author)

The manuscript by Wangm Shao et al. shows an interesting system made of chiral Au-nanoparticles coated with achiral organic molecules. Thanks to the plasmonic bands, the systems reach remarkable circularly polarized photo and electroluminescence (CPPL/CPEL). In a few cases this strong response is due to circular self-extinction/scattering. The topic is relevant, given the strong interest for chiral materials and CP-OLEDs devices, the results are significant and mostly well-rationalized, so that I think that the manuscript could fit in Nature Communications. On the other hand, there are major points that need to be fully clarified. For this reason, I think that a thoroughly revised version of the manuscript should be reconsidered before a final decision.

- 1) The authors write that in ir reference system employing achiral coated Au-nanosphere the CD/CPL signal is very weak. If I understand, this system has no element of chirality, therefore its chiroptical properties (CD, CPL, etc.) should be exactly zero. There is no possibility of chiral excitons within an achiral material. If the CD signal is significantly non-zero, there is clearly an issue that should be identified and fixed. Please note that I consider this point a dealbreaker.
- 2) I find figure 1 and 2 very confusing. It is not clear which system each spectrum corresponds to (different concentration of the nanoparticles, TDBC, ratios). For example, in what do the 3 panels of Fig 1d and 1e differ? Moreover, only the CD in mdeg is shown, I do not understand the reason of the left scale (g_{ext} , which is not shown in those graphs). Similar considerations apply for Fig. S9-11.
- 3) The authors define the various g -factors correctly and consistent with the literature of the field, then they report and discuss percentage g -factors. I do not understand the reason for this choice and I would advise using the standard definition.
- 4) The maximum g_{el} reported with lanthanides is actually 1 with EQE approximately 0.05% (see 10.1002/adfm.201603719), with different lanthanide complexes 0.48% was achieved (10.1039/D1TC05023K). Please revise the text.
- 5) The effect on circular polarization of reflection onto the cathode has been investigated in 10.1002/adfm.201603719.
- 6) The structural/morphological differences between helicoid iii and iv should be better explained.
- 7) Concerning the relationship between polarization an efficiency of a device, the authors may be interested in this recent review 10.1002/adma.202406550 by Li et al.
- 8) TDBC should be defined.

Reviewer #4

(Remarks to the Author)

The work entitled "Circularly polarized OLEDs constructed from chiral plasmonic nanoparticle–molecule hybrids" by

Jianpeng Zheng et.al. reports on the development of CP-OLEDs by addressing the balance between large dissymmetry (g_{EL}) factor and external quantum efficiency (EQE). The authors utilized chiral plasmonic NPs based upon helicoidal morphologies to construct CP-OLEDs, to achieve high EQE of 2.4% and g_{EL} factor of 0.31.

CP-OLEDs utilizing chiral plasmonic NPs based upon helicoidal morphologies is not new. For example, in a past publication (from the manuscript references 15 – 17), by a randomly dispersing the chiral NPs, a g_{EL} of 0.2 is reported. There are publications already showing EQE around 2.5% and g_{EL} factor approximately around 0.32 utilizing different chiral constructs (<https://www.nature.com/articles/s41467-022-35699-z>, <https://www.chinesechemsoc.org/doi/full/10.31635/ccschem.023.202303346>). So in terms of EQE and g_{EL} , this is not a substantial improvement.

Even though the authors had fabricated chiral plasmonic NPs based upon helicoidal morphologies to realize such reported EQE and g_{EL} factors, it seems to be an improvement in particular morphology based CP-OLEDs. There is no novelty or significant optical background mechanism involved in this work. So, it doesn't match the publication standard of Nature Communications. Detailed optical studies are not conducted to discuss various plasmonic properties and lifetime/stability of the device.

In conclusion, the manuscript should be published in different journal (Communication Chemistry or Communication Materials). It is not suitable for publication in Nature Communications.

Further critical questions need to be addressed in the work:

In figure 1(d, e) and related supporting information figure(s), the influence of Chiral plasmon & chiral plasmon-chiral exciton has been discussed. However, there is no detailed optical modes (dark/bright) discussion for such standalone plasmonic structures versus Plasmonic + TBDC structures.

How about the optical properties involving tuning/detuning of upper/lower excitons for a chiral plasmonic NPs versus TBDC coupled chiral plasmonic NPs (differences in term of plasmon & exciton).

As author discuss about the significance of plasmonic enhancement as one of the important parameter in achieving high EQE and g_{EL} , the optical background study directly influencing EQE (fig 4b) is not clear! Discussion on how the near-field properties fare better in their plasmonic constructs are needed!

Supp. Fig. 21 color scale shows only +- distribution which can't be a parameter to estimate the plasmonic enhancement. It can reveal the plasmonic modes but not the enhancement factor.

For the overshoot effect based results, how about the recovery time (for a multiple continuous on/off cases, as seen from Supp. Fig. 27b, d). Furthermore, as seen from the time-dependent EL, the tail signature remains ~ 0.3 seconds! In such situation, how about a device facing continuous on-off state and recovery? How it can affect the life-time stability such CP-OLED device?

Reviewer #5

(Remarks to the Author)

Version 1:

Reviewer comments:

Reviewer #1

(Remarks to the Author)

After careful evaluation of revised manuscript, I believe that the manuscript can be suitable for publication in journal "Nature Communications" in presented form. All necessary additions and clarifications are made in the revised manuscript.

Reviewer #2

(Remarks to the Author)

Reviewer's Report for revised manuscript:

"Circularly polarized OLEDs constructed from chiral plasmonic nanoparticle–molecule hybrids"

By: Jiapeng Zheng, Yuang Fu, Jing Wang, Wei Zhang, Xinhui Lu, Hai-Qing Lin, Lei Shao & Jianfang Wang

As far as I am concerned, the manuscript may be published in the present form, since the Authors have correctly addressed all the issues I have raised.

Reviewer #3

(Remarks to the Author)

I have carefully considered the revisions made by the authors. The manuscript has been improved and the critical aspects have been addressed by the authors. For this reason, I think that the manuscript can now be accepted in Nat. Comm.

Reviewer #4

(Remarks to the Author)

I appreciate the authors' efforts in this study and acknowledge the significance of achieving a high external quantum efficiency (EQE) alongside a gEL value of 0.31. I agree with the authors that while previous works have reported larger EQEs, they have often demonstrated relatively low gEL values, making this work noteworthy for presenting a balanced improvement in both metrics. Furthermore, the application of chiral plasmonic nanoparticles (NPs) in enhancing circularly polarized OLED (CP-OLED) properties has yielded better results compared to prior studies, particularly in terms of gEL.

However, this manuscript does not meet the high standards required for publication in Nature Communications. The fabrication of chiral plasmonic nanoparticles described in this work is not novel and has been thoroughly explored in previous publications. Additionally, while the reported gEL value of 0.31 is an improvement, it is not sufficient as a standalone merit for publication in a journal of this caliber. From the perspective of plasmonic physics, the underlying novelty and scientific insights provided by this study are insufficient to warrant publication.

In summary, while this work makes incremental improvements in gEL along with EQE for CP-OLEDs, it does not offer significant advancements in the field of plasmonic physics or present transformative ideas that align with the standards of Nature Communications. Therefore, I recommend rejection of this manuscript.

Reviewer #5

(Remarks to the Author)

Text coding: *black italic, reviewers' comments*; black normal, authors' response; **purple normal, changes made in the manuscript files**

Response to Reviewer #1

Comments: Wang *et al.* report on the assembly of chiral plasmonic nanoparticles (NPs) and supramolecular aggregates to construct various chiral polymer organic light-emitting diodes (CP-OLEDs). The multiscale chirality transfer, plasmonic enhancement, and suppression of the overshoot effect in the aggregates containing chiral plasmonic nanoparticles and chromophores within the emissive layer yield an external quantum efficiency (EQE) of 2.5% and a gEL factor of 0.31. The work demonstrates that chiral plasmonic nanoparticles (NPs) can serve as promising candidates for ultracompact chiral photoelectric devices, presenting a novel concept worth publishing. However, several issues have been identified in the manuscript, as outlined below. I recommend that the paper be reevaluated once these problems have been addressed.

Response: We thank the reviewer for the effort on evaluating our work, the highly positive comment, and the insightful questions and suggestions. The comments and suggestions from this reviewer are very helpful in improving the quality of our work.

Question 1: The authors mention the multiscale chirality transfer from chiral plasmonic nanoparticles to chiral plasmonic nanoparticles, but what is the medium of chiral transmission? The authors should be more deeply discussed.

Response: We thank the reviewer for the insightful question on the multiscale chirality transfer from the chiral plasmonic nanoparticles to the nanoparticle–molecule hybrids. We have explored two ways of chirality transfer that contribute to the chiral emissions of the (chiral plasmonic NP)–molecule hybrids, as shown in the figure below. The first one is the transfer of geometric chirality from the 432 helicoid IV NPs to the supramolecular aggregates of TDBC molecules assembled around the NP. Similar to the way that twisted elastic substrates transfer geometric chirality to nanoparticle chains conformally coated on them (*Nat. Mater.* **2016**, *15*, 461), the 432 helicoid IV NPs can direct the chiral stacking of TDBC aggregates. The formed supramolecular aggregates support chiral excitons with their chiral emissions exhibiting a bisignate spectral shape. The second way of chirality transfer contributing to chiral emissions is through chiral plasmon resonance. Previous works (*Adv. Mater.* **2023**, *35*, 2210477) have demonstrated that gammadion-shaped nanostructures can transfer their structural chirality to unpolarized light through optical field modulation, inducing circularly polarized luminescence (CPL) from achiral nano-emitters. In our work, the strong chiroptical response of the plasmonic 432 helicoid III NPs enables the circular-polarization filtering of unpolarized emitted photons through circular-polarization-preferential scattering and absorption by the chiral NPs.

Caption | Multiscale chirality transfer enabling the chiral emissions of the (chiral plasmonic NP)-molecule hybrid films. **a–c** Transfer of geometric chirality from 432 helicoid IV to the supramolecular aggregates of TDBC molecules assembled around the NP. The 432 helicoid IV NPs can direct the chiral stacking of TDBC aggregates to generate chiral excitons that exhibit CPL. **d–f** Transfer of chirality to emission polarization through chiral plasmon resonance. The 432 helicoid III NPs with strong chiral plasmon response can filter unpolarized emitted photons through circular-polarization-preferential scattering and absorption.

We have added the illustrations of multiscale chirality transfer in the revised Fig. 1 (part c) and provided the discussion about this point in the middle of the paragraph right before Fig. 1.

“..... We focused on two critical mechanisms (Fig. 1c) and designed various CP-OLED architectures (Fig. 1d). The first is the transfer of the geometric chirality from NPs to molecule aggregates¹⁹. Chiral NPs can direct the chiral stacking of supramolecular aggregates to generate chiral Frenkel excitons. This chiral-exciton-dominated CP-OLED-1 device was fabricated based on 432 helicoid IV–molecule hybrids, showing the bisignate CPEL with $|g_{\text{EL}}|$ of ~0.15 and EQE of 1.3%. The second is the transfer of optical chirality²⁰. The strong chiroptical response of the plasmonic 432 helicoid III and nanotriskelion NPs enables the circular-polarization filtering of unpolarized emissions through circular-polarization-preferential scattering and absorption by the chiral NPs. This chiral-plasmon-dominated CP-OLED-2 device was fabricated based on 432 helicoid III–molecule hybrids, showing a higher EQE value (2.5%) and a larger $|g_{\text{EL}}|$ factor (0.31).....”

The reference (*Adv. Mater.* **2023**, 35, 2210477) has been added.

20. Mendoza-Carreño, J. *et al.* Nanoimprinted 2D-chiral perovskite nanocrystal metasurfaces for circularly polarized photoluminescence. *Adv. Mater.* **35**, 2210477 (2023).

Question 2: Please provide a detailed explanation about the overshoot effect. Does it affect the *GEL* factor of OLED device?

Response:

We really appreciate the question raised by this reviewer. Our response includes the following two points.

(1) Previous works have demonstrated that the luminance overshoot can be explained by the radiative recombination of pre-stored electrons and injected holes (*Chem. Phys. Lett.* **2004**, 397, 87; *Mater. Horiz.* **2021**, 8, 2785). Our experimental results on the overshoot effect, including the time-dependent luminating photographs and transient EL profiles are summarized in Supplementary Fig. 22a,c in the updated Supplementary Information. We plotted a schematic illustrating the mechanism of the overshoot effect (Supplementary Fig. 22d). A detailed explanation on the overshoot effect is as follows. In the Ref-OLED device without any metal NPs, TDBC monomers have two imidazole rings and show electron-donating properties. Upon the application of a pseudo-rectangular voltage pulse to OLED, the luminance intensity increases, overshoot to a maximum value, and thereafter decreases and reaches a steady value. The luminance overshoot can be explained by the radiative recombination of pre-stored electrons and injected holes. After a few rounds of electrical switching, some electrons are pre-trapped in the TDBC aggregates. The new round of electrical switching triggers the radiative recombination of injected holes and the pre-stored electrons, producing a spike with a super-high intensity at the transient EL rising edge (Supplementary Fig. 22d-i). When the pre-stored electrons are dissipated, the luminance decreases until it reaches a steady state, and newly injected electrons start to accumulate in the TDBC aggregates (Supplementary Fig. 22d-ii). As a comparison, we also measured the transient EL spectra and recorded the images of the transient luminescence process of the CP-OLED-2 devices, which used the hybrids of TDBC aggregates and the chiral Au NPs with excellent electrical conductivity. The amount of pre-stored electrons can be significantly reduced because of the existence of the conductive Au NPs, thereby eliminating the overshoot effect. Compared with Ref-OLED, the measured CP-OLED-2 device underwent a relatively slow rise in its luminance intensity upon the application of a voltage, which took about 0.2 s until the luminance intensity reached a steady state. The slower rise of the luminance intensity in CP-OLED-2 compared with that of its Ref-OLED counterpart results from both the lack of pre-stored carriers and the larger emissive layer thickness of the CP-OLED-2 devices (640 ± 50 nm *versus* 560 ± 37 nm). After the application of a voltage pulse, the transport and recombination of charge carriers requires longer time.

We have updated Supplementary Fig. 22 in the Supplementary Information. We have also added a detailed explanation about the overshoot effect in the caption of Supplementary Fig. 22 as well as in the main text.

Supplementary Fig. 22 | Comparison of the Ref-OLED and CP-OLED-2 devices in terms of the overshoot effect. **a** Time-dependent luminating photographs of a Ref-OLED device and a CP-OLED-2 device after a voltage of $U = 6$ V was applied. **b** Normalized EL intensity changes over time for a Ref-OLED device and a CP-OLED-2 device under the application of $U = 4$ V. The CP-OLED-2 device with metal NPs exhibits a higher stability because of the suppression of the overshoot effect. **c** Transient EL profiles of a Ref-OLED device under the application of $U = 4$ V. **d** Schematic illustrating the overshoot phenomenon in the Ref-OLED and CP-OLED-3 devices. The luminance first increases and overshoots to a maximum value (i) before decreasing to a steady value (ii). TDBC monomers have two imidazole rings (Fig. 1c) and show electron-donating properties. In the Ref-OLED device, after a few rounds of electrical switching, some electrons were pre-stored in the TDBC aggregates. The new round of electrical switching triggers the radiative recombination of the injected holes and the pre-stored electrons (i), producing a spike with a super-high intensity at the transient EL rising edge⁷. After the pre-stored electrons are consumed because of recombination with injected holes, the luminance decreases until it reaches a steady value, and new electrons start to accumulate in the TDBC aggregates again (ii). A recovery time of 0.1–0.2 s is necessary for the EL of Ref-OLED to reach its steady state. The overshoot effect can also be observed in the CP-OLED-3 devices using separate TDBC and Au NP layers. **e** Transient EL profile of a CP-OLED-2 device under the application of $U = 4$ V. **f** Schematic

illustrating the suppression of the overshoot effect in the CP-OLED-1 and CP-OLED-2 devices. The CP-OLED-1 and CP-OLED-2 devices used the hybrids of TDBC aggregates and the chiral Au NPs with excellent electrical conductivity. The amount of pre-stored electrons can be significantly reduced, thereby eliminating the overshoot effect. Compared with Ref-OLED, the measured CP-OLED-2 device underwent a relatively slow rise in its luminance intensity upon the application of a voltage, which took about 0.2 s until the luminance intensity reached a steady state. The slower rise of the luminance intensity in the CP-OLED-2 devices compared with that of their Ref-OLED counterparts results from both the lack of pre-stored carriers and the larger emissive layer thickness of the CP-OLED-2 devices (640 ± 50 nm versus 560 ± 37 nm). After the application of a voltage pulse, the transport and recombination of charge carriers requires longer time.

The detailed explanation on the overshoot effect has been added at the end of the paragraph right before Fig. 5 in the main text.

“..... The overshoot effect came from the radiative recombination of pre-stored electrons and injected holes, characterized by a spike with a super-high intensity at the transient EL rising edge (Supplementary Fig. 22)^{46,47}. We found that the overshoot effect can make the OLEDs work in an unstable state, reducing the EQE and luminous lifespan (Supplementary Fig. 22b and Fig. 23). The transient EL examination of the CP-OLED-2 devices have revealed that the chiral NPs with excellent electrical conductivity is helpful to significantly reduce the amount of pre-stored electrons, eliminating the overshoot effect (Supplementary Fig. 22).”

The references (*Chem. Phys. Lett.* **2004**, 397, 87; *Mater. Horiz.* **2021**, 8, 2785) have been added in the main text.

46. Ma, C. W. *et al.* Time-resolved transient electroluminescence measurements of emission from DCM-doped Alq₃ layers. *Chem. Phys. Lett.* **397**, 87–90 (2004).

47. Chen, J. *et al.* An unprecedented spike of the electroluminescence turn-on transience from guest-doped OLEDs with strong electron-donating abilities of host carbazole groups. *Mater. Horiz.* **8**, 2785–2796 (2021).

(2) We have also explored the effect of the overshoot effect on the CPEL performance. The overshoot effect reduces the EQE and luminous lifespan because of the accumulation of electrons and holes inside the emissive layer, as shown in the revised Supplementary Fig. 22b and 23. Such negative effects have been observed in the CP-OLED-3 devices that used separate TDBC and chiral Au nanotriskelion layers. The separation between the Au NPs and TDBC prevents the quick conduction of pre-stored electrons in TDBC. As a result, unprecedented spikes of the turn-on EL were observed under $U > 6$ V. We found that the overshoot effect also leads to a degraded g_{EL} -factor, as shown in the revised Fig. 5 in the main text and Supplementary Figs. 31–33. The performance degradation of the emission circular polarization dissymmetry may result from molecule deterioration. To avoid the negative effect of the overshoot effect, the g_{EL} spectra of CP-OLED-3 working at low voltages (< 6 V) were used to measure the CPEL performance of the CP-OLED-3 devices.

We have added the discussion on the effect of the overshoot effect on the OLED device g_{EL} factor at the beginning of the paragraph right after Fig. 5 in the main text.

“In addition, as U became larger, the CP-OLED-2 devices exhibited gradually increasing $|g_{EL}|$ that tended to reach saturation (Fig. 5c, Supplementary Figs. 27–29), while $|g_{EL}|$ of CP-OLED-3 reached its maximum value before suffering from an overshoot-effect-induced degradation (Fig. 5d, Supplementary Figs. 30–32). The different $|g_{EL}|$ – U evolution behaviors imply

Question 3: *The device preparation process should be described in detail. For example, how the emissive layer is prepared, what solvent is used, and what is the concentration of spirochaeta. How is the uniformity of the emissive layer directly dropped on the ZnO surface and whether it affects the effect of the OLED device?*

Response: We thank the reviewer for asking the experimental details. We are sorry for that we did not provide this information in our original manuscript. We have therefore added a detailed description of the preparation process of the OLED devices in the revised manuscript. The preparation process is illustrated in the updated Supplementary Fig. 20a and also mentioned in the main text. We have also described the device preparation process in detail in Supplementary Methods in the Supplementary Information. Information on the uniformity of the emissive layers and their influence on the performance of the OLED devices will be answered in our response to Question 7 below.

The added Supplementary Fig. 20 is as follows.

Supplementary Fig. 20 | Fabrication and characterization of the OLED devices. a

Schematics illustrating the preparation process of the OLED devices. The photographs are also provided (bottom right corner) to show a typical OLED device at the different preparation stages, including (i) the structure of FTO/ZnO/emissive layer/BCP, (ii) the structure of FTO/ZnO/emissive layer/BCP/Ag, and (iii) the OLED device after connection to the circuit. **b** Energy level diagram of the OLED using TDDBC aggregates as the emissive layer³. ZnO and BCP were selected as the electron and hole transport layer, respectively. **c** SEM images of the emissive layer in the L-OLED-2 device, including the image of the (L-432 helicoid III NP)–(TDDBC aggregate) hybrid film surface (top), the magnified SEM images demonstrating that the L-432 helicoid III NPs were embedded in the TDDBC aggregates (bottom, left), the cross-section image of the L-OLED-2 device before the deposition of BCP and Ag, revealing the FTO, ZnO, and emissive layer (bottom, right). **d** SEM images of the emissive layer in the Ref-OLED device, including the image of the TDDBC aggregate hybrid film surface (top), the magnified SEM images demonstrating the high uniformity of the TDDBC film (bottom, left), the cross-section image of the Ref-OLED device before the deposition of BCP and Ag, revealing the FTO, ZnO, and emissive layer (bottom, right). The thickness of the TDDBC layer in Ref-OLED is 560 ± 37 nm. The (432 helicoid III NP)–(TDDBC aggregate) hybrid film in CP-OLED-2 showed a larger thickness of 640 ± 50 nm because of the large size of the 432 helicoid III NPs.

The device fabrication details have been added in Supplementary Methods in the Supplementary Information as follows.

OLED fabrication. The proof-of-concept OLED devices were fabricated using an inverted device structure composed of fluorine-doped tin oxide (FTO), zinc oxide (ZnO), the emissive layer, bathocuproine (BCP), and a silver layer. ZnO and BCP were selected as the electron and hole transport layers, respectively^{3,4}. The preparation process of the OLED devices followed the steps shown in Supplementary Fig. 20a. The FTO strip electrode of 2 mm in width was washed by sequential ultrasonication in detergent, deionized water, acetone, and isopropanol for 20 min each, and dried on a hot plate before being cleaned by ultraviolet ozone for 20 min. After being dried, the FTO electrode was spin-coated with the ZnO precursor solution at 2,000 rpm for 30 s and then annealed at 200 °C for 30 min to obtain the electron transport layer. The ZnO precursor solution was prepared by adding $\text{Zn}(\text{CH}_3\text{COO})_2 \cdot 2\text{H}_2\text{O}$ (100 mg) and ethanolamine (28.29 μL) into 2-methoxyethanol (973 μL) and stirred overnight before use. The ZnO layer was then covered by adhesive tapes, to form a channel of 12 mm in length and 4 mm in width on top of the FTO electrode. To prepare the Ref-OLED device, TDBC aqueous solution (0.92 mM, 80 μL) was drop-cast onto the ZnO layer. The substrate was spun at a low speed of 300 rpm and sequentially placed onto a hot plate at 45 °C. The TDBC aggregate film with high uniformity was produced after water evaporation, forming an emissive layer with a thickness of 560 ± 37 nm. To prepare the emissive layers of the CP-OLED-1 and CP-OLED-2 devices, the TDBC aqueous solution (0.92 mM, 80 μL) was first mixed with the 432 helicoid IV NP solution (100 μL) and the 432 helicoid III NP solution (50 μL) before the drop-casting process, respectively. To prepare the emissive layer of the CP-OLED-3 device, the Au nanotriskelions were first deposited on the FTO/ZnO electrode by drop-casting to form a layer of a high-density random NP array. The TDBC aggregate film was then prepared separately on top of the NPs following the same procedure of preparing the emissive layer of Ref-OLED. On top of the prepared emissive layer, the hole transport layer was deposited by drop-casting the BCP solution under rotation at a speed of 500 rpm and a sequential drying process. The BCP solution was prepared by dissolving BCP (0.1 g) in isopropyl alcohol (40 mL) under stirring for 24 h. The devices were then stored in a glove box with nitrogen environment and then evaporated with Ag electrodes of 2 mm in width through a shadow mask.

We have also revised the first paragraph of the section “Chiral-exciton-dominated CP-OLEDs” in the main text to include a brief introduction of the details of device preparation.

“We fabricated proof-of-concept CP-OLED devices using an inverted device structure composed of fluorine-doped tin oxide (FTO), zinc oxide (ZnO), the emissive layer, bathocuproine (BCP), and the Ag layer. (see Fig. 4, Supplementary Fig. 20, and Supplementary Methods)^{39,40}. We measured the CPEL spectra, transient electroluminescence (EL) spectra, and EQEs of the devices and characterized their circular polarized emission capability with the g_{EL} factor (Supplementary Figs. 21–23). g_{EL} has the same definition as g_{PL} , i.e., $g_{\text{EL}} = 2(I_{\text{LCP}}^{\text{E}} - I_{\text{RCP}}^{\text{E}})/(I_{\text{LCP}}^{\text{E}} + I_{\text{RCP}}^{\text{E}})$, where $I_{\text{LCP}}^{\text{E}}$ and $I_{\text{RCP}}^{\text{E}}$ are the LCP and RCP EL intensities, respectively. The (432 helicoid IV NP)–(TDBC aggregate) hybrid film was

Question 4: *More importantly, why the authors use this device structure rather than the commonly used ITO/PEDOT:PSS/HTL/EL/ETL/LiF/Al structure type? How did the authors consider it?*

Response: We thank the reviewer for this insightful question! We had indeed considered the ITO/PEDOT:PSS/HTL/EL/ETL/LiF/Al configuration during the fabrication of the CP-OLED devices. Previous works have proved that TDBC-based OLED devices with the conventional structure can be fabricated using poly(9,9'-dioctylfluorene-co-N-(4-butylphenyl)diphenylamine) (TFB) as the hole transport layer (HTL) (*Adv. Opt. Mater.* **2013**, *1*, 503). However, additional fabrication steps are required to deposit the emissive layer containing colloidal Au NPs, which are dispersed in aqueous solutions, onto the hydrophobic TFB layer. We therefore the hydrophilic ZnO layer as the electron transport layer and employed the inverted device structure for fabrication simplicity. In our work, the fabricated CP-OLED devices only involve the design of the emissive layer. Our material should also work for the structure of ITO/PEDOT: PSS/HTL/EL/ETL/LiF/Al, although the device fabrication requires additional steps such as the stabilization of the hybrids of Au NPs and emissive molecules in an organic solvent.

We have added the related discussion in the first paragraph of the Discussion section.

“..... with outstanding circular polarization emission dissymmetry and efficiencies. The proposed method only involves the design of the emissive layer and does not require any specialized changes on the OLED device architecture or molecular modification on the chromophore. Our strategy is therefore compatible with the current manufacturing technology of OLEDs and can also be applied to devices with different structures³⁹. In addition, the capability of chiral plasmonic NPs in”

Question 5: *Although the authors give a corresponding explanation for g_{EL} larger than g_{PL} , this is still unreasonable for the structure of electroluminescent devices. Whether it is chiral organics, complexes, or nanoclusters that are used as emissive layers, the g_{EL} of the device is usually smaller than the g_{PL} , the accepted reason is that the chirality will be reversed during the reflection process so that the g_{EL} signals counteract each other. Why is it?*

Response: We thank the reviewer for this insightful question! For the three different types of CP-OLEDs introduced in our work, we will discuss the comparison of g_{EL} and g_{PL} of the emissive layer in each device separately as a response. In short, our answer includes two points. First, our CP-OLED-1 devices have an emissive layer supporting chiral excitons and their excitation by charge injection is much more efficient than by photon excitation. Despite the chirality reversion of the emissions introduced by electrode reflection, the reflected backward emissions are attenuated by the plasmonic NPs embedded in the emissive layer, and therefore cannot compete with the circular polarization of the forward emissions. The resultant overall g_{EL} is still distinctly higher than g_{PL} . Second, both the CP-OLED-2 and CP-OLED-3 devices employed Au NPs with strong chiroptical responses as circular-polarization filters. Both forward emissions and reflected backward emissions are circularly polarized by the chiral NPs to carry the same handedness, preventing any circular polarization cancellation and boosting the asymmetric EL. The resultant overall g_{EL} is higher than g_{PL} . The explanations for all the three types of CP-OLED devices are further provided below in detail.

(1) In the CP-OLED-1 devices, the (432 helicoid IV NP)–(TDBC aggregate) hybrid film was employed as the emissive layer. We believe the much larger value of g_{EL} than that of g_{PL} should be attributed to the more efficient generation of chiral excitons through charge injection, as shown in the revised Supplementary Fig. 25. When the hybrid is excited optically, the chromophores far away from the plasmonic NPs make a remarkable yet almost achiral contribution to the PL signal. In comparison, in the electrically excited hybrid, charges will preferentially inject into the molecule aggregates on the metal NP surface because of the high electrical conductivity of Au, giving rise to a much more efficient generation of chiral excitons than achiral excitons. In the CP-OLED-1 devices, molecules supporting achiral excitons far away from the metal NPs can be viewed as in parallel connection with the (metal NP–molecule) hybrid. Due to the high electrical conductivity of the metal NPs, the current injected into the (metal NP–molecule) hybrid is much larger than that injected into the molecules supporting achiral excitons. In addition, we have observed that g_{EL} further increases with increasing applied voltage U in the CP-OLED-1 device (Supplementary Fig. 25b), leading to an even higher g_{EL} . The plasmonic near-field of the metal NPs accelerates the radiative emission of chiral excitons near the NP surface because of the Purcell effect. Chiral excitons near the NP surface therefore decay radiatively in a faster manner than that of achiral excitons far away from the NPs. As the applied voltage U is enlarged to a critical value U_{sa} , the number of electrically excited achiral excitons reaches a saturated value, i.e., all achiral excitons are excited simultaneously in the radiative lifetime of achiral excitons. Further increasing U will not increase the number of achiral excitons. In contrast, excited chiral excitons near the NP surface can still become enriched when U is larger than U_{sa} , since chiral excitons near the plasmonic surface decay faster. The superior enrichment of chiral excitons at high voltages gives rise to superior CPEL of the OLED device, i.e., larger g_{EL} . We also performed the transient CPEL measurements on the D-OLED-1 devices at high voltages (Supplementary Fig. 25c). The CPEL spectra show gradually enlarged splitting between the LCP and RCP EL peaks as charges are accumulated in the TDBC aggregates, indicating that the increase of injected electron–hole pairs results in more efficient generation of chiral excitons than that of achiral excitons. One should also note that an enlarged emission g_{PL} factor can be achieved by increasing the amount of TDBC molecules in the NP–molecule hybrid, since the molecules assembled around the chiral NPs are excited by light more efficiently because of the enhanced plasmonic near-field, which leads to the increase of the amount ratio between chiral excitons and achiral excitons (Supplementary Fig. 25a), similar to the results when the applied voltage is increased to generate CPEL.

We have revised Supplementary Fig. 25 and added the above discussion on g_{EL} and g_{PL} in the caption of Supplementary Fig. 25.

Supplementary Fig. 25 | Comparison between CPPL and CPEL measurements on the (432 helicoid IV NP)–(TDBC aggregate) hybrids. **a** CPPL measurement on the (D-432 helicoid IV NP)–(TDBC aggregate) hybrids. The increased TDBC amount caused the gradual enrichment of chiral excitons, leading to the increased CPPL intensity. **b** CPEL measurement on the CP-OLED-1 device employing the (D-432 helicoid III NP)–(TDBC aggregate) hybrid film as the emissive layer. The comparison among the CPEL measurements on the same device working at different voltages suggests that as the applied voltage U is enlarged, excited chiral excitons near the NP surface become enriched while the EL from achiral excitons reaches saturation, giving rise to superior CPEL of the OLED device working at high voltages. **c** Transient CPEL profiles of the D-OLED-1 device after the 8 V voltage was applied to the device. The CPEL spectra were obtained at an exposure time of 0.01 s. The CPEL intensity increased from ~100 counts to ~500 counts, with a gradually enlarged splitting between the LCP and RCP EL peaks. Such results indicate that the increase of injected electron–hole pairs can guide the generation of chiral excitons with increased numbers.

We have also added the above discussion in the paragraph right before Fig. 4 in the main text.

“We believe the much larger value of electroluminescent g_{EL} than that of fluorescent g_{PL} , when the same (432 helicoid IV NP)–(TDBC aggregate) hybrid films are employed as the emissive layer, should be attributed to the more efficient generation of chiral excitons through charge injection. When the hybrids are excited optically, the chromophores far away from the plasmonic NPs make a remarkable yet almost achiral contribution to the PL signal. In comparison, in the electrically excited hybrids, charges will preferentially inject into the molecule aggregates on the metal NP surface because of the high electron conductivity of Au,

giving rise to a much more efficient generation of chiral excitons than achiral excitons. The molecules supporting achiral excitons far away from the metal NPs can be viewed as in parallel connection with the (metal NP–molecule) hybrids. Because of the high electron conductivity of the metal NPs, the current injected into the (metal NP–molecule) hybrids is much larger than that injected into the molecules supporting achiral excitons. Despite the chirality reversion of the emissions introduced by electrode reflection⁸, the backward-reflected emissions are attenuated by the plasmonic NPs embedded in the emissive layer. The resultant overall g_{EL} is therefore distinctly higher than g_{PL} . In addition, we have observed that g_{EL} further increases with increasing applied voltage U in the CP-OLED-1 device (Supplementary Fig. 25). Owing to the plasmonic Purcell effect, chiral excitons near the NP surface radiatively decay in a faster manner than that of achiral excitons far away from the NPs. As U is enlarged to a critical value U_{sa} , the number of electrically excited achiral excitons reaches a saturated value, i.e., all achiral excitons are excited simultaneously in the radiative lifetime of achiral excitons. Further increasing U will not increase the number of achiral excitons. In contrast, excited chiral excitons near the NP surface can still become enriched when U is larger than U_{sa} , since chiral excitons near the metal surface decay faster. The superior enrichment of chiral excitons at high voltages gives rise to distinguished CPEL performances. We also performed the transient CPEL measurement and observed a gradually enlarged splitting between the LCP and RCP EL peaks (Supplementary Fig. 25), indicating that the increase of injected electron–hole pairs can guide the generation of chiral excitons with increased numbers.”

(2) In the CP-OLED-2 devices, the (432 helicoid III NP)–(TDBC aggregate) hybrid film was employed as the emissive layer. The strong chiroptical response of the plasmonic 432 helicoid III NPs enables the circular-polarization filtering of unpolarized emissions through circular-polarization-preferential scattering and absorption by the chiral NPs. Previous works have demonstrated that the EL emits forward to the transparent FTO electrode or emits backward and gets reflected by the Ag back electrode (*Adv. Funct. Mater.* **2017**, *27*, 1603719; *Nat. Photonics* **2023**, *17*, 193). Unlike the emissive layers mentioned by the reviewer, the local emitting center in our work, i.e., TDBC molecules, in CP-OLED-2 is not inherently in a strong circular polarization state. Instead, both the forward and backward-reflected emissions are circularly polarized by the chiral NPs to carry the same handedness, preventing any circular polarization cancellation and boosting the asymmetric EL. The CP-OLED-2 device using the L-432 helicoid III NPs with g_{ext} of -0.08 at 600 nm shows $g_{EL} = +0.13$. We further measured the CPPL spectra of the two samples based on the same 432 helicoid III NPs, as shown in the revised Supplementary Fig. 26. The first sample is the (432 helicoid III NP)–(TDBC aggregate) hybrid film. Its PL light path is similar to that of the forward emission in the CP-OLED-2 device. The sample shows a $g_{PL} = +0.02$ at 600 nm. The second sample is the TDBC aggregate film assembled with a separate layer of L-432 helicoid III, whose PL light path is similar to that of the backward-reflected emissions in the CP-OLED-2 device. The sample shows a $g_{PL} = +0.12$ at 600 nm. We therefore confirmed that the high g_{EL} -factor is originated from both forward and backward-reflected emissions.

We have revised Supplementary Fig. 26 and added the above discussion on g_{EL} and g_{PL} in the caption of Supplementary Fig. 26.

Supplementary Fig. 26 | Comparison between CPPL and CPEL measurements on the (432 helicoid III NP)-(TDBC aggregate) hybrids. **a** Schematic illustrating the CPEL emissions in the CP-OLED-2 device. Previous works have demonstrated that the EL emits forward to the transparent FTO electrode or emits backward and gets reflected by the Ag back electrode⁸. Both the forward and backward-reflected emissions can be circularly polarized by the chiral NPs to carry the same handedness, preventing any circular polarization cancellation and significantly boosting the asymmetric EL. **b** The CP-OLED-2 device using the L-432 helicoid III NPs with g_{ext} of -0.08 at 600 nm exhibits $g_{\text{EL}} = +0.13$. The SEM image and g_{ext} spectrum of the L-432 helicoid III NPs are shown in Supplementary Fig. 28a,b. **c,d** CPPL emission in the (432 helicoid III NP)-(TDBC aggregate) hybrid film, mimicking the forward emissions in the CP-OLED-2 device and showing a $g_{\text{PL}} = +0.02$ at 600 nm. **e,f** CPPL emissions in the TDBC aggregate film assembled with a separate layer of L-432 helicoid III, mimicking the backward emissions in the CP-OLED-2 device and showing a $g_{\text{PL}} = +0.12$ at 600 nm. These experimental results confirmed that the g_{EL} -factor in CP-OLED-2 was originated from both the forward and backward-reflected emissions.

(3) In the CP-OLED-3 devices, the emissive layer was obtained by assembling a chromophore film and a layer of aligned nanotriskelions that were prepared separately. Both the forward and reflective EL from the TDBC layer were circularly polarized by the nanotriskelion layer. Only the plasmonic polarization-filtering effect contributes to the device CPEL. We also measured the CPPL spectra of separately assembled TDBC and Au nanotriskelion layers, as shown in revised Supplementary Fig. 27. The L-nanotriskelions with g_{ext} of $+0.025$ at 600 nm support a CPPL emission with $g_{\text{PL}} = -0.07$. The same L-nanotriskelions was used in the CP-OLED-3 devices, leading to the CPEL emissions of $g_{\text{EL}} = -0.08$ at the same wavelength. The similar values of g_{PL} and g_{EL} imply that only the plasmonic polarization-filtering effect contributes to the device CPEL.

We have revised Supplementary Fig. 27 and added the discussion on the g_{EL} and g_{PL} in the caption of Supplementary Fig. 27.

Supplementary Fig. 27 | Comparison between CPPL and CPEL measurements of TDBC films assembled with a layer of aligned chiral Au nanotriskelions. **a** Photograph of a transparent flat substrate deposited with the aligned Au nanotriskelions. **b** SEM image of the Au nanotriskelion layer in **(a)**. The inset shows the magnified SEM image of the sample. The chiral NPs lie on the flat substrate and are aligned in an up-and-down direction. **c** g_{ext} spectrum of the Au nanotriskelions in **(b)**. **d** Schematic illustrating the CPPL measurement of the TDBC film assembled with a layer of chiral Au nanotriskelions. **e, f** CPPL spectra (**e**, top), PL spectra (**e**, bottom), and g_{PL} spectra (**f**) of the TDBC film assembled with the aligned chiral Au nanotriskelion layer. **g** Schematic illustrating the CPEL emissions in the CP-OLED-3 device. The EL emits forward to the transparent FTO electrode or emits backward and gets reflected by the Ag back electrode. All the emitted unpolarized photons are circularly polarized by the nanotriskelions. **h** CPEL spectra of the CP-OLED-3 device using the same Au nanotriskelions layer in **(b, c)**. **i** g_{EL} spectrum of the CP-OLED-3 device. The L-nanotriskelions with $g_{\text{ext}} = +0.025$ at 600 nm support CPPL emissions with $g_{\text{PL}} = -0.07$. The same L-nanotriskelions was used in the CP-OLED-3 device, leading to CPEL emissions of $g_{\text{EL}} = -0.08$ at the same wavelength. The similar g -factors demonstrated that only the plasmonic polarization-filtering effect contributes to the device CPEL.

We have also added the discussion about the comparison between g_{EL} and g_{PL} for the CP-OLED-2 and 3 devices in the paragraph right before the “Luminous efficiency of the CP-OLEDs” section.

In all CP-OLED-2 and CP-OLED-3 devices, the EL emitted forward to the transparent fluorine-doped tin oxide (FTO) electrode or emitted backward and got reflected by the Ag back electrode⁸. We demonstrated that both the forward and backward-reflected emissions were circularly polarized by the chiral NPs to carry the same handedness, preventing any circular polarization cancellation and significantly boosting the asymmetric EL (Supplementary Figs. 26 and 27)⁴³. We have also fabricated the CP-OLED-2 and CP-OLED-3 devices using the chiral NPs with different g_{ext} -factors (Supplementary Figs. 28–33). The resultant CPEL dissymmetry of the CP-OLED-2 and CP-OLED-3 devices increased almost linearly with $|g_{ext}|$ of the corresponding chiral NPs, reaching values higher than those of CP-OLED-1 with strong asymmetric emissions from chiral excitons (Fig. 5a).

Question 6: In the manuscript, the authors demonstrate that the CP-OLEDs exhibit efficient CP emission performance, achieving a high external quantum efficiency (EQE) of 2.5% and a considerable g_{EL} factor of 0.31, as compared to the limited selection of CPEL devices presented in Supplementary Table 1. However, the authors have only compared a small number of CPEL devices, and the performance of the CP-OLEDs in this study does not appear particularly exceptional. Therefore, a more thorough discussion of the results is warranted.

Response: We thank this reviewer for the great suggestion! We have followed the reviewer’s suggestion by including the performances of more previously reported CP-OLED devices in revised Supplementary Table 1 (see below). To facilitate reading, we further plotted the $|g_{EL}|$ factors and EQEs in revised Fig. 5e, which clearly shows the exceptional performance of the CP-OLEDs in our study. Previously reported CP-OLEDs generally present the inverse relationship between the EQE value and $|g_{EL}|$ factor. In contrast, the CP-OLEDs in our work can give both high EQE and $|g_{EL}|$. The EQEs of our devices are about two orders of magnitude higher than those of lanthanide-complex-based devices and the $|g_{EL}|$ factors are about two orders of magnitude larger than those of chiral-cluster-based devices.

Supplementary Table 1 | Comparison of different CP-OLED devices^{5,6}

Emissive layer	g_{EL}	$ g_{EL} $	EQE _{max} (%)	Reference
CP-OLED-1	~0.15	~0.15	1.3	This work
CP-OLED-2	0.31	0.31	2.5	This work
CP-OLED-3	0.32	0.32	2.1	This work
30 wt% S-BACzBO:PPF	-5.3×10^{-4}	0.00053	36.8	Mater. Horiz. 11 , 1752 (2024)
M,M-RBNN	$+1.91 \times 10^{-3}$	0.00191	36.6	Adv. Mater. 36 , 2307420 (2024)
P-BN[9]H	-6.2×10^{-3}	0.0062	35.4	Angew. Chem. Int. Ed. 63 , e202401835 (2024)

Address : SHATIN · NT · HONG KONG

Tel 電話 : (852) 3943 6339 / 3943 6154

E-mail 電郵 : physics@cuhk.edu.hk

地 址 : 香 港 · 新 界 · 沙 田

Fax 傳真 : (852) 2603 5204

URL 網址 : http://www.phy.cuhk.edu.hk/

R-CzOBN:POT2 T:BN1	$+2.8 \times 10^{-3}$	0.0028	33.2	Adv. Mater. 34 , 2109147 (2022)
(R)-Czp- tBuCzB	$+1.54 \times 10^{-3}$	0.00154	32.1	Angew. Chem. Int. Ed. 62 , e202217045 (2023)
TCTA:R- TRZOBN:Ir1	$+3.2 \times 10^{-3}$	0.0032	32	Adv. Funct. Mater. 33 , 2215179 (2023)
IrR	$+2.6 \times 10^{-3}$	0.0026	31.9	J. Mater. Chem. C 12 , 3997 (2024)
(R)-OBN-Cz	2.30×10^{-3}	0.0023	31.7	Adv. Mater. 31 , 1900524 (2019)
(M)-helicene- BN	-2.2×10^{-3}	0.0022	30.7	CCS Chem. 4 , 3463 (2022)
R-NPACZ: (tfmppy) ₂ Ir(po p)	$+2.2 \times 10^{-3}$	0.0022	30.7	Adv. Mater. 36 , 2311857 (2024)
(P)-BN-Py	-4.37×10^{-4}	0.000437	30.6	Adv. Mater. 35 , 2305125 (2023)
Λ -Ir-(R- camphor)	-4.36×10^{-4}	0.000436	30.5	Adv. Funct. Mater. 31 , 2102898 (2021)
(+)-(S)-ax- DMAC	-2.0×10^{-3}	0.002	30.1	Chem. Eng. J. 468 , 143508 (2023)
(R)-OBN-2CN- BN	$+1.43 \times 10^{-3}$	0.00143	29.4	Adv. Mater. 33 , 2100652 (2021)
3 wt% Ir(tptpy) ₂ acac:(S)-BNPCN-p- CP	-2.32×10^{-3}	0.00232	29.3	Angew. Chem. Int. Ed. 62 , e202300492 (2023)
(R)-ODQPXZ	-6.4×10^{-4}	0.00064	29.3	Adv. Opt. Mater. 9 , 2100017 (2021)
D-(R)- BPSPXZ	-8.5×10^{-3}	0.0085	28.5	Adv. Opt. Mater. 12 , 2302730 (2024)
(M)-DB-O	$+2.2 \times 10^{-3}$	0.0022	27.5	Adv. Mater. 36 , 2308314 (2024)
R-DOBNT	-1×10^{-3}	0.001	25.6	Adv. Mater. 34 , 2204253 (2022)
(S)-SCFPY	$+3.6 \times 10^{-3}$	0.0036	23.3	Sci. China Chem. 65 , 1347 (2022)
(S)-SFOT	$+1.0 \times 10^{-3}$	0.001	23.1	J. Am. Chem. Soc. 142 , 17756 (2020)
(R,R)- pTpAcBP	-1.0×10^{-3}	0.001	22.1	Angew. Chem. Int. Ed. 60 , 23619 (2021)
R/S- (BINAP) ₂ Cu(μ - I ₂)	3.0×10^{-3}	0.003	21.7	J. Mater. Chem. C 11 , 1329 (2023)
(S,S)-CPAD	-1.4×10^{-3}	0.0014	21.5	Angew. Chem. Int. Ed. 60 , 20728 (2021)
R-Ax-CN	$+4.2 \times 10^{-3}$	0.0042	21	Chem. Eng. J. 462 , 142123 (2023)
(S,S)-TpAc- TRZ (undoped devices)	$+1.5 \times 10^{-3}$	0.0015	20.7	Adv. Funct. Mater. 31 , 2106418 (2021)
(+)-BN4	$+4.6 \times 10^{-4}$	0.00046	20.6	Adv. Mater. 34 , 2105080 (2022)

(R)-SPOCN+(S)-OSFSO	$+2.5 \times 10^{-3}$	0.0025	20.4	Angew. Chem. Int. Ed. 61 , e202200290 (2022)
(R)-TRZ-MeIAc	$+6.4 \times 10^{-4}$	0.00064	20.3	Mater. Horiz. 8 , 547 (2021)
(M,M)-CNSPZ	$+2.9 \times 10^{-3}$	0.0029	20.03	Adv. Opt. Mater. 11 , 2203030 (2023)
(S)-OSFSO	$+3.1 \times 10^{-3}$	0.0031	20	Angew. Chem. Int. Ed. 60 , 8435 (2021)
(+)-(S,S)-CAI-Cz	-1.7×10^{-3}	0.0017	19.7	Angew. Chem. Int. Ed. 57 , 2889 (2018)
(R)-OBN-3CN	-1.1×10^{-3}	0.0011	19.7	Chem. Eng. J. 476 , 146511 (2023)
R-OBN-AICz	$+4.7 \times 10^{-4}$	0.00047	19	J. Mater. Chem. C 10 , 4805 (2022)
S-PXZ-PT	-1.3×10^{-3}	0.0013	18.5	Mater. Horiz. 8 , 3417 (2021)
(R)-P-BPCZ4	-5.2×10^{-3}	0.0052	18.3	Adv. Opt. Mater. 9 , 2100596 (2021)
(S)-BPPOACZ	$+4.5 \times 10^{-3}$	0.0045	17.8	Adv. Sci. 7 , 2000804 (2020)
(+)-(R,R)-MC	$+1.5 \times 10^{-3}$	0.0015	17.1	CCS Chem. 4 , 3540 (2022)
(R)-BN-2Mcp:Ir(mppy) ₃	-1.3×10^{-3}	0.0013	17.1	Chem. Commun. 59 , 1473 (2023)
S-OBS-Cz	-1×10^{-3}	0.001	17	ACS Appl. Mater. Interfaces 13 , 56413 (2021)
P-Pt	-1.6×10^{-3}	0.0016	16.26	Chem. Eur. J. 25 , 5672 (2019)
S-P	$+1.6 \times 10^{-3}$	0.0016	15.8	ACS Appl. Mater. Interfaces 14 , 1578 (2022)
λ -Ir(dfppy) ₂ (S-sdpp)	-2.1×10^{-3}	0.0021	14.6	J. Mater. Chem. C 9 , 5244 (2021)
(P)-QAO-PhCz	$+1.5 \times 10^{-3}$	0.0015	14	Chem. Commun. 57 , 11041 (2021)
R _p -MAC*-Cu-CzP	$+4.5 \times 10^{-4}$	0.00045	13.2	Sci. China Chem. 66 , 2274 (2023)
TAPC:R-TRZ	$+7.25 \times 10^{-3}$	0.00725	12.7	Adv. Opt. Mater. 10 , 2201793 (2022)
(-)-(S)-Cz-Ax-CN	-1.2×10^{-2}	0.012	12.5	Angew. Chem. Int. Ed. 59 , 3500 (2020)
(S)-OBN-tBuCz	$+1.57 \times 10^{-3}$	0.00157	12.4	Acta Chim. Sinica 79 , 1401 (2021)
S-o-BAMCN	$+6 \times 10^{-4}$	0.0006	12.4	Adv. Funct. Mater. 31 , 2103875 (2021)
R-OBN-DPA	$+2.3 \times 10^{-3}$	0.0023	12.3	J. Mater. Chem. C 7 , 7045 (2019)
R-pSACODP	-1.1×10^{-3}	0.0011	12	ChemPhotoChem 8 , e202300253 (2024)
D-(R)-C-DpCpN-Trz	$+7.6 \times 10^{-4}$	0.00076	11.3	Adv. Funct. Mater. 2314205 (2024)

R-Pt	0.06	0.06	11.27	Adv. Opt. Mater. 8 , 2000775 (2020)
(M)-QPO-PhCz	$+1.6 \times 10^{-3}$	0.0016	10.6	J. Mater. Chem. C 10 , 4393 (2022)
(R)-BIPNX-TRZ	-8.4×10^{-4}	0.00084	10.2	Chem. Eng. J. 471 144709 (2023)
S-CPDCz	-3.7×10^{-3}	0.0037	10.1	J. Mater. Chem. C 7 , 14511 (2019)
R-Ag ₆ (PTLT) ₆	-5.3×10^{-3}	0.0053	10	Nano Res. 16 , 7733 (2023)
S-BN-CF	0.026	0.026	9.3	Adv. Funct. Mater. 28 , 1800051 (2018)
S _p -5	$+4.3 \times 10^{-3}$	0.0043	7.8	ACS Appl. Mater. Interfaces 13 , 25186 (2021)
(R)-CO-PhDPA	-2.4×10^{-3}	0.0024	6.4	J. Org. Chem. 89 , 3605 (2024)
R-P	$+1.6 \times 10^{-3}$	0.0016	6.2	Adv. Opt. Mater. 11 , 2300550 (2023)
D-(S,S)-DCz	-8.3×10^{-4}	0.00083	5.5	ACS Appl. Mater. Interfaces 12 , 23172 (2020)
(S-2Cz) _{0.2} -(PFpy) _{0.8} -(Ir(MDQ) ₂) _{0.1}	-0.014	0.014	4.1	Adv. Funct. Mater. 33 , 2309133 (2023)
(Λ)-Ir ₂ ((-)-CS)/((-)-IL	-2.5×10^{-3}	0.0025	4.09	Angew. Chem. Int. Ed. 62 , e202302160 (2023)
R-5 (undoped devices)	$+5.6 \times 10^{-3}$	0.0056	2.79	Chem. Commun. 55 , 9845 (2019)
L-HP-NTi	-0.023	0.023	2.3	Adv. Mater. 35 , 2209495 (2023)
(R)-BP2	$+1.11 \times 10^{-3}$	0.00111	2.15	ACS Appl. Mater. Interfaces 12 , 9520 (2020)
[Sm(tta) ₃ (d-phen)]	0.011	0.011	1.55	J. Mater. Chem. C 11 , 1265 (2023)
(R)/(S)-[tmd]	3.0×10^{-3}	0.003	1.4	Inorg. Chem. 60 , 13557 (2021)
R-3	-9.8×10^{-3}	0.0098	1.2	J. Mater. Chem. C 8 , 15669 (2020)
R-P37+f8bt	-0.02	0.02	1.08	Adv. Opt. Mater. 12 , 2301513 (2023)
S-BN-tCz	2.13×10^{-3}	0.00213	1	Chem. Eur. J. 27 , 589 (2021)
(R)-C'3	1.0×10^{-3}	0.001	0.8	Adv. Funct. Mater. 30 , 2004838 (2020)
(S-M) _{0.2} -(BP) _{0.8}	0.01	0.01	0.69	Angew. Chem. Int. Ed. 62 , e202214424 (2023)
F8BT+S-6	-1.86×10^{-2}	0.0186	0.54	ACS Appl. Mater. Interfaces 13 , 55420 (2021)
(S-P2) _{0.6} -(NPy) _{0.4}	$+4.6 \times 10^{-2}$	0.046	0.21	Angew. Chem. Int. Ed. 61 , e202202718 (2022)

$M\text{-H6(TMS)}_2$	8.0×10^{-3}	0.008	0.21	Chem. Sci. 12 , 5522 (2021)
CsEu((-)-hfbc) ₄	-1	1	0.05	Adv. Funct. Mater. 27 , 1603719 (2017)
Ln(III) complexes	0.51	0.51	0.48	J. Mater. Chem. C 10 , 463–468 (2022)
CsEu((+)-hfbc) ₄	0.73	0.73	0.0042	Adv. Mater. 27 , 1791 (2015)
CsEu((-)-hfbc) ₄	-0.61	0.61	0.002	Adv. Funct. Mater. 27 , 1603719 (2017)

Fig. 5 | Circular polarization dissymmetry and energy efficiency performance of the CP-OLEDs with different configurations. e Comparison of various CP-OLEDs in terms of EQE value and $|g_{EL}|$ factor. The data of previously reported CP-OLEDs were summarized in Supplementary Table 1. Different from previously reported CP-OLEDs showing the inverse relationship between the EQE value and $|g_{EL}|$ factor, the CP-OLEDs constructed from the chiral plasmonic NP–molecule hybrids exhibit both high EQE values and large $|g_{EL}|$ factors.

We have also added the discussion in the paragraph right before the Discussion section as follows.

“..... Previously reported CP-OLEDs generally present the inverse relationship between the EQE value and $|g_{EL}|$ factor (Supplementary Table 1)^{7,48,49}. For example, lanthanide complex-based CP-OLEDs show extremely high $|g_{EL}|$ factors but low EQE values, while chiral cluster-based CP-OLEDs generally exhibit high EQE values and low $|g_{EL}|$ factors. In contrast, the CP-OLEDs in our work can achieve both high EQE and $|g_{EL}|$ values. The EQEs of our devices are about two orders of magnitude higher than those of lanthanide-complex-based devices and the $|g_{EL}|$ factors are about two orders of magnitude larger than those of chiral-cluster-based devices.”

Question 7: I observe that the chiral plasmonic NPs possess > 100 nm size, and what is the thickness of the CP-OLED emissive layer? Will such an emission layer be detrimental to device performance?

Response: We have measured the thickness and uniformity of the emissive layers. We have added the results in revised Supplementary Fig. 20c and d. The thickness of the TDBC layer

in Ref-OLED is 560 ± 37 nm, and the (432 helicoid III NP)–(TDBC aggregate) hybrid film in CP-OLED-2 shows a larger thickness of 640 ± 50 nm because of the larger size of the 432 helicoid III NPs. We have performed the transient EL measurements to analyze the influence of such an emission layer on device performance. A previous work showed that OLED devices fabricated with TDBC layers (thickness: 20 nm) exhibited a rise time of $0.2 \mu\text{s}$ (*Adv. Opt. Mater.* **2013**, *1*, 503). As shown in Supplementary Fig. 20, the emissive layers in our CP-OLED devices are thicker. The transport and recombination of charge carriers require a longer time to reach the steady state. As a result, our CP-OLED devices present a longer rise time (about $200 \mu\text{s}$) upon the application of a voltage pulse. Despite the slower response, the emissive layers with large thicknesses exhibit an excellent CPEL performance, as shown in Supplementary Figs. 28–30. We therefore conclude that our thick emissive layers containing the plasmonic metal NPs will not be detrimental to the CP-OLED device performance.

Supplementary Fig. 20 | Fabrication and characterization of the OLED devices. c SEM images of the emissive layer in the L-OLED-2 device, including the image of the (L-432 helicoid III NP)–(TDBC aggregate) hybrid film surface (top), the magnified SEM images demonstrating that the L-432 helicoid III NPs were embedded in the TDBC aggregates (bottom, left), the cross-section image of the L-OLED-2 device before the deposition of BCP and Ag, revealing the FTO, ZnO, and emissive layer (bottom, right). **d** SEM images of the emissive layer in the Ref-OLED device, including the image of the TDBC aggregate hybrid film surface (top), the magnified SEM images demonstrating the high uniformity of the TDBC film (bottom, left), the cross-section image of the Ref-OLED device before the deposition of BCP and Ag, revealing the FTO, ZnO, and emissive layer (bottom, right). The thickness of the TDBC layer in Ref-OLED is 560 ± 37 nm. The (432 helicoid III NP)–(TDBC aggregate) hybrid film in CP-OLED-2 showed a larger thickness of 640 ± 50 nm because of the large size of the 432 helicoid III NPs.

Question 8: The EQE of 2.5% for the OLED device is not competitive for measuring CPEL. As is widely recognized, testing CPEL presents significant challenges; therefore, the authors should provide detailed methodologies to ensure the accuracy of the test results.

Response: We thank this reviewer for this insightful suggestion. We have added the detailed methodology for CPEL measurements in revised Supplementary Figs. 21 and 23.

Supplementary Fig. 21 | Detailed methodologies for CPEL measurements. **a** Schematic illustrating the home-built optical setup for CPEL measurements. Two sets of circular polarized plates were employed. Each was composed of a quarter-wave plate (QWP) and an ultrathin linear polarizer (LP). These two plate sets were integrated into one bracket (see the photograph), one for measuring the LCP component and the other for recording the RCP component of the emission. **b** Schematic illustrating the process for the measurement of LCP and RCP components. Two voltage pulses with a 2-second interval were applied to accurately measure the intensity of CPEL components. During a 2-second interval, the plate groups were rapidly moved to switch the component measurement. Various OLED devices of the same type were examined to ensure the accuracy of the CPEL measurements. **c–f** CPEL performance of the Ref-OLED device. The TDBC aggregate film was used as the emissive layer (**c**). The transient profiles of the EL intensity were recorded when the LCP and RCP components of the EL were selected (**d**). Unprecedented overshoot spikes of the EL turn-on transience appeared when a voltage of $U = 6$ V was applied. The CPEL spectra were obtained in two cycles (**e**). We therefore calculated the electroluminescent g_{EL} spectra (**f**), demonstrating the achiral EL from the Ref-OLED device. The TDBC J-aggregate film shows a very weak chiroptical response and the Ref-OLEDs exhibited a nearly achiral EL feature. **g–j** CPEL performance of the CP-OLED-3 device. A combination of the TDBC aggregate film and L-nanotriskelions was used as the emissive layer (**g**). The transient profiles of the LCP and RCP EL intensity were recorded (**h**). The CPEL spectra were obtained in two cycles

(i). We therefore calculated the electroluminescent g_{EL} spectra (j), demonstrating the chiral EL emission from the CP-OLED-3 device. The CP-OLED-3 device displayed distinctly different LCP and RCP EL intensities.

Supplementary Fig. 23 | EQE measurements of the OLED devices. **a** Schematic showing the optical setup for the measurement of the external quantum efficiency (EQE) of an OLED device. The emitted EL light was collected by an optical fiber and directed to a spectrometer. The current was simultaneously recorded from the ammeter under the application of different voltages. **b** Absolute radiation spectra at varying applied voltages for a typical OLED device. **c–f** Voltage-dependent current density and EQE for a Ref-OLED device (**c**), a CP-OLED-1 device (**d**), a CP-OLED-2 device (**e**), and a CP-OLED-3 device (**f**), respectively.

We have also added the discussion about the details of CPEL measurements in Supplementary Methods.

CPEL measurement. CPEL spectroscopy was performed on a home-built optical setup. As shown in Supplementary Fig. 21, two sets of circular polarization plates were employed, each of which was composed of a quarter-wave plate (QWP, Union Optic Inc., 550–750 nm) and an ultrathin linear polarizer (LP, Union Optic Inc., 550–900 nm). These two plate groups were integrated into one bracket, one for measuring the LCP component and the other for recording the RCP component of the emission. The intensity of the emitted electroluminescence (EL) from the OLED device passing through the plate groups was measured by the spectrometer system composed of a CCD camera (Princeton Instruments, Pixis 400) and a monochromator (Acton, SpectraPro 2360i). The OLED devices were connected with a Source Meter instrument (Keithley 2636B), which provided voltage pulses. Voltage pulses with a 2-second interval were applied to accurately measure the intensity of CPEL components. During a 2-second interval, two plate groups were rapidly moved to switch the LCP and RCP component measurement. We can therefore record the LCP and RCP components under different voltage pulses. Various OLED devices of the same type were examined to ensure the accuracy of the CPEL measurements (Supplementary Fig. 21).

We first prepared Ref-OLED devices employing the TDBC J-aggregate film alone as the emissive layer. The TDBC J-aggregate film showed a weak chiroptical response and the Ref-OLEDs exhibited a nearly achiral EL feature. Under two cycles of CPEL measurement, we recorded the transient profiles of the EL intensity, CPEL spectra, and g_{EL} spectra, demonstrating the unpolarized EL emitted from the Ref-OLED device. We also measured the CPEL profiles from our CP-OLED devices. For example, the CP-OLED-3 devices displayed distinctly different LCP and RCP EL intensities.

Response to Reviewer #2

Comments: *This is a nice report of chiroptical and electro-chiroptical properties of hybrid systems consisting chiral Au nanoparticles and of 5,6-dichloro-2-[[5,6-dichloro-1-ethyl-3-(4-sulfobutyl)-benzimidazol-2-ylidene]-propenyl]-1-ethyl-3-(4-sulfobutyl)-benzimidazolium hydroxide (TDBC) chromophore molecule. The manuscript reports on an exquisitely novel research possibly impacting on CD spectroscopy, sensing and photoluminescence. It deals with the interaction of plasmonic CD/CPPL/CPEL from Au NPs and of excitonic CD/CPPL/CPEL from TDBC. My approval to publication is quite likely, subject to the Authors' responses to the following questions/comments.*

Response: We thank this reviewer for the effort on evaluating our work, the highly positive comments, and the insightful questions and suggestions. The comments and suggestions from this reviewer are very helpful in improving the quality of our work.

Question 1: *Is TDBC a non-chiral molecule? If so, is excitonic CD coming just from chiral arrangements around the chiral NPs? It is the experience of this Reviewer that chirality of the organic molecule is a bonus for the production of CD spectra.*

Response: TDBC is a non-chiral molecule. The CD signal in the (432 helicoid IV)–(TDBC aggregate) hybrid films comes from the chiral arrangement of the molecule aggregates around the chiral NPs. We believe that the 432 helicoid IV NPs can serve as scaffolds that trigger the chiral arrangement of TDBC monomers and therefore promote the generation of high-quality chiral excitons in the TDBC aggregates. As the figure below shows (the same data are also given in Fig. 3 in the main text and Supplementary Figs. 4 and 15 of our updated manuscript), in the absence of any metal NPs, achiral TDBC monomers can self-assemble into aggregates with random local chiral configurations that endow the aggregates with a very weak ensemble chiroptical response. The control samples of (Au nanosphere)–(TDBC chiral aggregate) hybrid films showed very weak extinction CD signals, with the intensities being one to two orders of magnitude smaller than those of the (432 helicoid IV NP)–(TDBC aggregate) hybrids. Note that both the 432 helicoid IV NPs and the Au nanospheres exhibited very weak far-field chiroptical response. Our results strongly support the argument that the chiral NPs can direct the chiral stacking of TDBC aggregates, and as a result generate chiral excitons in the formed supramolecular chromophores.

Caption | CD spectra of different TDBC aggregate films. **a** Pure TDBC aggregate film forming on a flat substrate. **b** (Au nanosphere)–(TDBC aggregate) hybrid film. **c** (432 helicoid IV)–(TDBC aggregate) hybrid film.

We have added the discussion about the chiroptical response of TDBC monomers and TDBC aggregates in the first paragraph under the “Fabrication of chiral plasmonic NP–molecule hybrids” section.

“We first prepared TDBC-based layer-by-layer J-aggregate films that support delocalized Frenkel excitons through evaporation-driven self-assembly (Supplementary Figs. 4 and 5)^{21,22}. Different from commonly used fluorescent dyes such as Rhodamine (Supplementary Fig. 6), achiral TDBC monomers can self-assemble into aggregates with random chiral configurations under solvent evaporation^{23,24}, endowing the aggregate film with a very weak ensemble chiroptical response. The formed chiral J-aggregate films exhibit

Question 2: *It is the experience of this Reviewer to define the circularly polarized emission as CPL (=circularly polarized luminescence) rather than CPPL (=circularly polarized photo-luminescence). Why did the Authors find it necessary to add an additional P to the mostly used CPL effect? Is CPPL an unknown effect to this Reviewer?*

Response: We thank the reviewer for raising this point. It is true that the CPL, i.e., circularly polarized luminescence, usually refers to circularly polarized photoluminescence in many references. Such circularly polarized photoluminescence spectra are usually measured from a circularly polarized luminescence spectrometer (e.g. JASCO CPL-300 in our work). In this context, the CPPL we used in our manuscript has the same meaning of the “CPL” mentioned by the reviewer and there are not any additional unknown effects. Nonetheless, literally, the meaning of the word “luminescence” includes both photoluminescence and electroluminescence. As pointed out by a recent review paper on CP-OLEDs (*Chem. Soc. Rev.* **2020**, *49*, 1331), which states “according to the different excitation mechanisms, CPL can be classified into CP photoluminescence (CPPL) and CP electroluminescence (CPEL)”, researchers not working in the field of “circular polarized luminescence” and general public readers may find it confusing if we use “CPL” referring to “circular polarized photoluminescence” and “CPEL” referring to “circular polarized electroluminescence”. We therefore would like to keep using the abbreviation of CPPL when referring to circular polarized photoluminescence.

To avoid any possible ambiguity, we have added the discussion in the first paragraph of the “Chiral-exciton-dominated CPPL in the 432 helicoid IV–molecule hybrids” section.

“Because of the correlation between the absorption CD and circularly polarized luminescence (CPL) emission of chiral molecules (Fig. 3c)^{25,26}, the (chiral plasmonic NP)–molecule hybrids are promising for developing emitters with improved CPL performance. The CPL performance of the emitters can be characterized by both CPPL and CPEL measurements. We first examined the CPPL responses of ……”

Question 3: *If CPPL is the same as CPL, due to Kasha’s rule, then only the longest wavelength component of the “excitonic” CPL is active, as in part one may see from Figure 13 and following in Supplementary Material. (See the numerous review articles, part of which are even cited in this manuscript.)*

Response: We thank the reviewer for this insightful question. Kasha’s rule is one of the crucial guiding principles describing how luminophores relax to the ground state (S_0) from an excited state (S_n). In many chiral luminescent materials, only the longest wavelength component of the “excitonic” circularly polarized luminescence (CPL) is active according to Kasha’s rule. However, there are several exceptions in chiral aggregates that can break Kasha’s rule. Previous works have proved that anti-Kasha’s rule can be observed in some chiral aggregates with tubular shapes, which show bisignate CPL signals associated with bisignate CD signals (*Angew. Chem. Int. Ed.* **2023**, *62*, e202212724). A theoretical work (*Chem. Sci.* **2024**, *15*, 16103) has also reported the bisignated CPL signal, which reveals significant anti-Kasha emissions, in cyanine aggregates. The weak electrostatic interaction in the aggregates results in an exciton splitting smaller than the thermal energy at room temperature. Two excited states can therefore be significantly populated according to a Boltzmann probability distribution, yielding the distinctive anti-Kasha bisignated CPL signal. The same mechanism can explain the bisignate profiles of both the CD and CPL of the (432 helicoid IV NP)–(TDBC aggregate) hybrid film observed in our work. We thereby believe that the 432 helicoid IV NPs can serve as the scaffolds that trigger the chiral arrangement of TDBC monomers and as a result promote the generation of chiral excitons, finally causing anti-Kasha emissions, although we are currently restricted by our limited experimental capacity to confirm the geometrical chiral arrangement of TDBC aggregates and to examine the energy splitting of formed chiral excitons.

We have added the discussion about the anti-Kasha process in the second paragraph under the “Chiral-exciton-dominated CPL in the 432 helicoid IV–molecule hybrids” section.

The bisignated CPPL signal reveals an anti-Kasha emissive nature of the (432 helicoid IV NP)–(TDBC aggregate) hybrid. The same anti-Kasha emissive feature was also observed in chiral aggregates with tubular shapes and was explained as follows^{25,36}. The weak electrostatic interaction in the molecule aggregates results in an exciton energy splitting smaller than thermal energy at room temperature. Both excited states, i.e., S_+ and S_- , can therefore be significantly populated according to a Boltzmann probability distribution. The decay of S_+ and S_- emits photons with opposite helicities and as a result yields the distinctive anti-Kasha bisignated CPPL signal. The same mechanism can explain the bisignate profiles of both CD and CPPL of the (432 helicoid IV NP)–(TDBC aggregate) hybrid films observed in our experiments. We thereby believe that the 432 helicoid IV NPs serve as the scaffolds that trigger the chiral arrangement of TDBC monomers and as a result promote the generation of chiral Frenkel excitons, although additional experimental evidence is needed to directly

confirm the geometrical chiral arrangement of TDBC aggregates on the NP surface and to examine the energy splitting of formed chiral excitons in detail.

A reference (*Chem. Sci.* **2024**, *15*, 16103) has been added.

36. Bertocchi, F. *et al.* Chiroptical properties of cyanine aggregates: hierarchical modelling from monomers to bundles. *Chem. Sci.* **15**, 16103–16111 (2024).

Question 4: *A better description of the experimental chiroptical experiments should be provided in the Supplementary file (which apparatuses were employed, the spectroscopic resolution, the number of scans, etc.)*

Response: We thank the reviewer for this insightful suggestion. The experimental chiroptical measurements include CD, CPPL, and CPEL. The CD and CPPL spectra were recorded on a CD spectrophotometer (JASCO J-1500) and CPL spectrometer (JASCO CPL-300), respectively. The CPEL spectroscopy was performed on a home-built optical setup. We have added the detailed methodology for CPEL measurements in revised Supplementary Fig. 21. We have also added the discussion about the details of CPEL measurements in Supplementary Methods. Please refer to our response to Question 8 of the Reviewer #1 above.

Question 5: *Minor points: (1) in the text please write, in more formal way, “TDBC concentration” rather than in the more concise way, [TDBC]. (2) The sentence on page 4 in the text “The observed bisignate CD spectrum, known as the Cotton effect,…” is inappropriate, since Cotton effect is an old way of calling CD, either monosignate or bisignate: Aimé Auguste Cotton lived in the eighteen/nineteen centuries, much before any CD experimentation, when only the optical rotation was known and CD was deduced therefrom. (3) On Page 7: what do the Authors mean by “more chiral excitons”? (4) On page 11: what do the Authors mean by “CP degrees”? Do they mean: “CP intensity”?*

Response: We thank the reviewer for carefully reading and pointing out these minor issues. We really appreciate the reviewer’s kindness of sharing the knowledge. We have addressed the questions one by one as shown below.

(1) In our work, 50 μL of the chiral NPs and TDBC at a specific concentration (0.92 mM) were used to fabricate the (chiral plasmonic NP)–(TDBC aggregate) hybrid films. We feel it would be better if we used the amount of the substance in the film, instead of the molecule concentration when preparing the hybrid film, to describe the sample. We therefore switched to the TDBC amount n_{TDBC} for different samples in the revised figures and manuscript.

(2) We have deleted the statement about “Cotton effect” in the first paragraph of the “Fabrication of chiral plasmonic NP–molecule hybrids” section.

“…… and a Davydov splitting of 68 meV (Supplementary Fig. 4). The observed bisignate CD spectrum indicates the energy splitting between Frenkel excitons owing to interchromophoric interactions^{25,26}. Previous studies have demonstrated ……”

(3) The inappropriate expression “more chiral excitons” has been changed to “the number of chiral excitons” (in the middle of the first paragraph after Fig. 2).

“..... Our observation suggests that higher n_{TDBC} gives rise to an increase in the number of chiral excitons in the TDBC aggregates formed on the chiral NP surface. The energy splitting in

(4) The inappropriate expression “CP degrees” have been changed into “CPL performance” in the first paragraph under the “Chiral-exciton-dominated CPL in the 432 helicoid IV–molecule hybrids” section.

“Because of the correlation between the absorption CD and circularly polarized luminescence (CPL) emission of chiral molecules (Fig. 3c)^{25,26}, the (chiral plasmonic NP)–molecule hybrids are promising for developing emitters with improved CPL performance. The CPL performance of the emitters can be characterized by both CPPL and CPEL measurements. We first examined the CPPL responses of

Response to Reviewer #3

Comments: *The manuscript by Wang, Shao et al. shows an interesting system made of chiral Au-nanoparticles coated with achiral organic molecules. Thanks to the plasmonic bands, the systems reach remarkable circularly polarized photo and electroluminescence (CPPL/CPEL). In a few cases this strong response is due to circular self-extinction/scattering. The topic is relevant, given the strong interest for chiral materials and CP-OLEDs devices, the results are significant and mostly well-rationalized, so that I think that the manuscript could fit in Nature Communications. On the other hand, there are major points that need to be fully clarified. For this reason, I think that a thoroughly revised version of the manuscript should be reconsidered before a final decision.*

Response: We thank this reviewer for the effort on evaluating our work, the highly positive comments, and the insightful questions and suggestions.

Question 1: *The authors write that in a reference system employing achiral coated Au-nanosphere the CD/CPL signal is very weak. If I understand, this system has no element of chirality, therefore its chiroptical properties (CD, CPL, etc.) should be exactly zero. There is no possibility of chiral excitons within an achiral material. If the CD signal is significantly non-zero, there is clearly an issue that should be identified and fixed. Please note that I consider this point a dealbreaker.*

Response: We thank the reviewer for this insightful question. TDBC monomer is a non-chiral molecule. It can self-assemble into aggregates with chiral configurations of random handedness that endow the aggregate ensembles with a very weak chiroptical response. Our experimental results have shown that the formed J-aggregate films exhibit a bisignate absorption circular dichroism (CD) signal centered around the absorption band (~585 nm), as

shown in the figure below (please also refer to Supplementary Figs. 4 and 15). The CD response of TDBC aggregates has also been reported in a previous work (*ACS Photonics* **2021**, 8, 901). Therefore, the control sample (reference system) shown in the revised Supplementary Fig. 15 is composed of achiral Au nanospheres and chiral TDBC aggregates. This system exhibits very weak extinction CD signals. Similar experimental phenomena have also been observed in published works using other achiral plasmonic NPs, including Au nanorod–TDBC (*Nanoscale* **2021**, 13, 15812) and Au@Ag nanocuboid (*ACS Nano* **2021**, 15, 2292).

Caption | CD spectra of different TDBC films. **a** Pure TDBC aggregate film. **b** (Au nanosphere)–(TDBC aggregate) hybrid film.

We have added the discussion on the chiroptical response of TDBC monomer and TDBC aggregates in the first paragraph of the “Fabrication of chiral plasmonic NP–molecule hybrids” section.

“We first prepared TDBC-based layer-by-layer J-aggregate films that support delocalized Frenkel excitons through evaporation-driven self-assembly (Supplementary Figs. 4 and 5)^{21,22}. Different from commonly used fluorescent dyes such as Rhodamine (Supplementary Fig. 6), achiral TDBC monomers can self-assemble into aggregates with random chiral configurations under solvent evaporation^{23,24}, endowing the aggregate film with a very weak ensemble chiroptical response. The formed chiral J-aggregate films exhibit a narrow absorption band at ~585 nm, an emission peak at ~600 nm, a bisignate absorption circular dichroism (CD) signal centered around the absorption band, and a Davydov splitting of 68 meV (Supplementary Fig. 4). The observed bisignate CD spectrum indicates the energy splitting between Frenkel excitons owing to interchromophoric interactions^{25,26}. Previous studies have demonstrated that various (plasmonic NP)–aggregate hybrids supporting enhanced light–matter interactions, such as (achiral Au nanorod)–(chiral aggregate)^{27,28} and (chiral Au nanorod dimer)–(TDBC aggregate) systems²⁹, can modify the chiroptical properties of the hybrid systems, resulting in mode splitting in CD spectra. We hereby demonstrate that the coupling between the chiral NPs and chiral TDBC aggregates can result in richer phenomena, including enlarged CD splitting and enhanced bisignate CD signals. Chiral plasmonic NP–molecule hybrids were fabricated through the assembly of TDBC aggregates and the colloidal chiral plasmonic NPs. The chiral NPs were

The references (*ACS Photonics* **2021**, 8, 901; *Nanoscale* **2021**, 13, 15812; *ACS Nano* **2021**, 15, 2292) have been added.

23. Guo, J. Q. *et al.* Optical chirality in a strong coupling system with surface plasmons polaritons and chiral emitters. *ACS Photonics* **8**, 901–906 (2021).
27. Wu, F. *et al.* Plexcitonic optical chirality: Strong exciton–plasmon coupling in chiral J-aggregate-metal nanoparticle complexes. *ACS Nano* **15**, 2292–2300 (2021).
28. Guo, J. Q. *et al.* Diverse axial chiral assemblies of J-aggregates in plexcitonic nanoparticles. *Nanoscale* **13**, 15812–15818 (2021).

Question 2: *I find figure 1 and 2 very confusing. It is not clear which system each spectrum corresponds to (different concentration of the nanoparticles, TDBC, ratios). For example, in what do the 3 panels of Fig 1d and 1e differ? Moreover, only the CD in mdeg is shown, I do not understand the reason of the left scale (g_{ext} , which is not shown in those graphs). Similar considerations apply for Fig. S9-11.*

Response: We thank the reviewer for this careful suggestion. To avoid misunderstanding, we have revised original Figs. 1 and 2 as new Figs. 2 and 3, respectively. In the revised manuscript, Figs. 2a,b and 3b show the extinction CD spectra of the chiral plasmonic NPs in aqueous solutions (black dashed lines) and the corresponding hybrid films prepared by forming TDBC aggregates on the surface of the chiral plasmonic NPs (red lines). The SEM images and g_{ext} spectra of the chiral NPs were moved to the updated Supplementary Figs. 7 and 12. We can therefore observe the change of the CD signal when different plasmonic NPs were used from updated Figs. 2 and 3. Revised Supplementary Figs. 9–11 also show only the CD spectra of the chiral plasmonic NPs and different hybrid systems.

Question 3: *The authors define the various g -factors correctly and consistent with the literature of the field, then they report and discuss percentage g -factors. I do not understand the reason for this choice and I would advise using the standard definition.*

Response: We thank the reviewer for this suggestion. We have used the standard expression of the g -factors in the revised manuscripts and figures.

Question 4: *The maximum g_{el} reported with lanthanides is actually 1 with EQE approximately 0.05% (see 10.1002/adfm.201603719), with different lanthanide complexes 0.48% was achieved (10.1039/D1TC05023K). Please revise the text.*

Response: We thank the reviewer for this correction. We have revised the text in the first paragraph of the introduction part.

“..... For example, chiral heterometallic clusters have been implemented in CP-OLEDs⁵ with EQEs close to 21%, but their $|g_{EL}|$ factors are limited at the level of 10^{-4} . Lanthanide complexes can achieve an extremely high $|g_{EL}|$ of 0.5–1.0, but their EQEs are very low (0.05–0.48%)^{8,9}. Novel chiral organic emitters are therefore needed to achieve significant improvements both in EQE and $|g_{EL}|$.”

The references (10.1002/adfm.201603719; 10.1039/D1TC05023K) have been added.

8. Zinna, F. *et al.* Design of lanthanide-based OLEDs with remarkable circularly polarized electroluminescence. *Adv. Funct. Mater.* **27**, 1603719 (2017).
9. Zinna, F. *et al.* Modular chiral Eu(III) complexes for efficient circularly polarized OLEDs. *J. Mater. Chem. C* **10**, 463–468 (2022).

We have also revised the discussion in the paragraph right before Discussion section in the main text as follows.

“..... Previously reported CP-OLEDs generally present the inverse relationship between the EQE value and $|g_{EL}|$ factor (Supplementary Table 1)^{7,48,49}. For example, lanthanide complex-based CP-OLEDs show extremely high $|g_{EL}|$ factors but low EQE values, while chiral cluster-based CP-OLEDs generally exhibit high EQE values and low $|g_{EL}|$ factors. In contrast, the CP-OLEDs in our work can achieve both high EQE and $|g_{EL}|$ values. The EQEs of our devices are about two orders of magnitude higher than those of lanthanide-complex-based devices and the $|g_{EL}|$ factors are about two orders of magnitude larger than those of chiral-cluster-based devices.”

Question 5: *The effect on circular polarization of reflection onto the cathode has been investigated in 10.1002/adfm.201603719.*

Response: We thank the reviewer for sharing this useful information. We have added the mentioned reference in our revised manuscript and provided more discussion accordingly. Please also refer to our response to Question 5 raised by Reviewer #1 above.

The added reference is as follows.

8. Zinna, F. *et al.* Design of lanthanide-based OLEDs with remarkable circularly polarized electroluminescence. *Adv. Funct. Mater.* **27**, 1603719 (2017).

We have provided the discussion on the reflection in the CP-OLED-1 devices at the end of the paragraph right before Fig. 4 as follows.

“We believe the much larger value of electroluminescent g_{EL} than that of fluorescent g_{PL} , when the same (432 helicoid IV NP)–(TDBC aggregate) hybrid films are because of the high electron conductivity of Au, giving rise to a much more efficient generation of chiral excitons than achiral excitons. The molecules supporting achiral excitons far away from the metal NPs can be viewed as in parallel connection with the (metal NP–molecule) hybrids. Because of the high electron conductivity of the metal NPs, the current injected into the (metal NP–molecule) hybrids is much larger than that injected into the molecules supporting achiral excitons. Despite the chirality reversion of the emissions introduced by electrode reflection⁸, the backward-reflected emissions are attenuated by the plasmonic NPs embedded in the emissive layer. The resultant overall g_{EL} is”

We have also provided the discussion on the reflection in the CP-OLED-2 and CP-OLED-3 devices in the paragraph right before the “Luminous efficiency of the CP-OLEDs” section.

“In all CP-OLED-2 and CP-OLED-3 devices, the EL emitted forward to the transparent fluorine-doped tin oxide (FTO) electrode or emitted backward and got reflected by the Ag back electrode⁸. We demonstrated that both the forward and backward-reflected emissions were circularly polarized by the chiral NPs to carry the same handedness, preventing any circular polarization cancellation and significantly boosting the asymmetric EL (Supplementary Figs. 26 and 27)⁴³. We have also fabricated”

Question 6: *The structural/morphological differences between helicoid iii and iv should be better explained.*

Response: 432 helicoid III has a cubic geometry, with six faces exhibiting pinwheel-like patterns. In comparison, 432 helicoid IV shows a rhombic-dodecahedral shape. The arrows in the SEM images indicate twisted edges at different chiral surfaces (see updated Fig. 1a in the revised main text). In the L-type 432 helicoid III NPs, the fourfold edges rotating clockwise around the twisted center are observed along the $\langle 100 \rangle$ directions. In contrast, the L-type 432 helicoid IV nanocrystals show two surfaces with opposite chirality along the $\langle 111 \rangle$ and $\langle 100 \rangle$ directions, resulting in a smaller g -factor. We have added the SEM images of 432 helicoid III and IV in revised Fig. 1 and explained their structural/morphological differences in the caption.

Fig. 1 | Chiral plasmonic NPs in the emissive layer of CP-OLEDs. a Schematics and scanning electron microscopy (SEM) images of the chiral plasmonic NPs. The 432 helicoid IV NPs show rhombic-dodecahedral shapes. The 432 helicoid III NPs are of a cubic geometry, with the six faces exhibiting pinwheel-like patterns. The nanotriskelions exhibit the anisotropic geometric morphology with triskelion-shaped wrinkles on the top and bottom surfaces. The L- and D-type chiral NPs were obtained from L- and D-GSH, respectively. The

distorted edges of the chiral NPs are shown by the orange arrows. All scale bars share the same value. **b** TDBC monomer, schematically represented by a pink platelet. **c** Schematics illustrating the chirality transfer from the chiral plasmonic NPs to the emission of the CP-OLEDs, generating circularly polarized luminescence. The chiral emissions result from either the formation of chiral excitons or the modulation by chiral plasmon resonance. **d** Schematic diagrams illustrating the device structures of the CP-OLEDs in this work. The photographs at the bottom show a typical CP-OLED device and its electroluminescence (inset).

We have provided the discussion on the structural/morphological differences in the middle of the paragraph right before Fig. 1.

“..... The emerging chiral Au NPs with helicoidal morphologies, such as 432 helicoid III, 432 helicoid IV, and nanotriskelions, have shown enhanced chiral light–matter interactions^{16–18}. The 432 helicoid III NPs and Au nanotriskelions show strong chiroptical responses, coming from their chiral arms with fourfold rotational symmetry along the $\langle 100 \rangle$ directions and threefold rotational symmetry along the $\langle 111 \rangle$ directions, respectively. The 432 helicoid IV NPs’ chiral arms exhibit opposite chirality along the $\langle 100 \rangle$ and $\langle 111 \rangle$ directions, leading to a weak far-field chiroptical response (Fig. 1a, Supplementary Figs. 2 and 3).....”

Question 7: Concerning the relationship between polarization and efficiency of a device, the authors may be interested in this recent review 10.1002/adma.202406550 by Li et al.

Response: We thank the reviewer for sharing this reference. It is very nice that this relevant published article introduces the EQE value and g_{EL} factor of various CP-OLEDs. Some data from this article were used and summarized in Supplementary Table 1. We also plot the data from Supplementary Table 1 in new Fig. 5e to demonstrate the substantial improvement of our work in the CPEL performance. Please also refer to our response to Question 6 raised by Reviewer #1 above.

Fig. 5 | Circular polarization dissymmetry and energy efficiency performance of the CP-OLEDs with different configurations. e Comparison of various CP-OLEDs in terms of EQE value and $|g_{EL}|$ factor. The data of previously reported CP-OLEDs were summarized in Supplementary Table 1. Different from previously reported CP-OLEDs showing the inverse relationship between the EQE value and $|g_{EL}|$ factor, the CP-OLEDs constructed from the chiral plasmonic NP–molecule hybrids exhibit both high EQE values and large $|g_{EL}|$ factors.

The references (10.1002/adma.202406550) have been added.

48. Guo, C. H. *et al.* Chiral co-assembly with narrowband multi-resonance characteristics for high-performance circularly polarized organic light-emitting diodes. *Adv. Mater.* **36**, 2406550 (2024).

Question 8: *TDBC should be defined.*

Response: We have defined TDBC in the middle of the paragraph right before Fig. 1.

“..... leading to a weak far-field chiroptical response (Fig. 1a, Supplementary Figs. 2 and 3). We used 5,6-dichloro-2-[[5,6-dichloro-1-ethyl-3-(4-sulfobutyl)-benzimidazol-2-ylidene]-propenyl]-1-ethyl-3-(4-sulfobutyl)-benzimidazolium hydroxide (TDBC) as the chromophore molecule (Fig. 1b). We focused on two

Response to Reviewer #4

Comments: *The work entitled “Circularly polarized OLEDs constructed from chiral plasmonic nanoparticle–molecule hybrids” by Jianpeng Zheng et.al. reports on the development of CP-OLEDs by addressing the balance between large dissymmetry (g_{EL}) factor and external quantum efficiency (EQE). The authors utilized chiral plasmonic NPs based upon helicoidal morphologies to construct CP-OLEDs, to achieve high EQE of 2.5% and g_{EL} factor of 0.31.*

CP-OLEDs utilizing chiral plasmonic NPs based upon helicoidal morphologies is not new. For example, in a past publication (from the manuscript references 15 – 17), by a randomly dispersing the chiral NPs, a g_{EL} of 0.2 is reported.

There are publications already showing EQE around 2.5% and g_{EL} factor approximately around 0.32 utilizing different chiral constructs (<https://www.nature.com/articles/s41467-022-35699-z>, <https://www.chinesechemsoc.org/doi/full/10.31635/ccschem.023.202303346>). So in terms of EQE and g_{EL} , this is not a substantial improvement.

Even though the authors had fabricated chiral plasmonic NPs based upon helicoidal morphologies to realize such reported EQE and g_{EL} factors, it seems to be an improvement in particular morphology based CP-OLEDs. There is no novelty or significant optical background mechanism involved in this work. So, it doesn't match the publication standard of Nature Communications. Detailed optical studies are not conducted to discuss various plasmonic properties and lifetime/stability of the device.

In conclusion, the manuscript should be published in different journal (Communication Chemistry or Communication Materials). It is not suitable for publication in Nature Communications.

Response: We thank the reviewer for the effort on evaluating our work, the questions and suggestions. It is great that the reviewer provided 5 published references to support the statements that (a) “CP-OLEDs utilizing chiral plasmonic NPs based upon helicoidal morphologies is not new” and (b) “There are publications already showing EQE around 2.5%

and g_{EL} factor approximately around 0.32 So in terms of EQE and g_{EL} , this is not a substantial improvement.” We really appreciate the clues provided by the reviewer. To facilitate reading, the 5 references used by the reviewer to support her/his statements are

(R1) Reference 15 in the original manuscript: *Nature* **2018**, 556, 360.

(R2) Reference 16 in the original manuscript: *Nat. Commun.* **2023**, 14, 3783.

(R3) Reference 17 in the original manuscript: *Nat. Commun.* **2020**, 11, 263.

(R4) *Nat. Commun.* **2023**, 14, 81.

(R5) *CCS Chemistry* **2023**, 5, 2760.

We have therefore read all of the 5 papers and all of their supplementary materials very carefully to make sure that we have not missed any important information. (R1) and (R3) are research works on the synthesis of chiral plasmonic nanoparticles, both of which are from the research group of Professor Ki Tae Nam, who is the pioneer of employing wet-chemistry methods to grow chiral metal NPs. Unfortunately, we could not find any data on electroluminescence or CP-OLEDs in (R1) or (R3). (R2) is our own work published in 2023. It introduces a new methodology of synthesizing chiral plasmonic NPs. We confirm that there is NOT any data of electroluminescence or CP-OLEDs in (R2). (R4) is an excellent research work on a circularly polarized photoluminescence study of achiral plasmonic NPs decorated with chiral molecules. However, we could not find any data on electroluminescence or CP-OLEDs in (R4). In the end, (R5) is a nice review article on circularly polarized luminescence materials and their applications, including the application in CP-OLEDs. However, we have not found any examples of “CP-OLEDs utilizing chiral plasmonic NPs based upon helicoidal morphologies”. We therefore are really confused by the reviewer’s comments. It appears that the references, which the reviewers particularly picked up to support her/his core statements (a) and (b), do not contain the information mentioned by the reviewer. More details are listed below.

Reviewer’s statement (a): *CP-OLEDs utilizing chiral plasmonic NPs based upon helicoidal morphologies is not new. For example, in a past publication (from the manuscript references 15–17), by a randomly dispersing the chiral NPs, a g_{EL} of 0.2 is reported.*

Response: We have carefully read the 3 publications (*Nature* **2018**, 556, 360; *Nat. Commun.* **2023**, 14, 3783; *Nat. Commun.* **2020**, 11, 263). These publications show the synthesis of various chiral NPs including 432 helicoid III, 432 helicoid IV, and nanotriskelions, but do not involve any experimental results of “CP-OLEDs (or electroluminescence) utilizing chiral plasmonic NPs”. We could not find any data of the electroluminescent g -factor, i.e., “ g_{EL} ”, in these publications. The g -factor data of 0.2 refers to the extinction g -factor of 432 helicoid III (*Nature* **2018**, 556, 360), or the photoluminescence emission anisotropy factor of the nanotriskelion–fluorophore hybrid nanostructures excited by linearly polarized light (*Nat. Commun.* **2023**, 14, 3783). We guess that the reviewer might have mixed up the data on different asymmetric factors and finally draw the inaccurate conclusion. To our knowledge, the strategy of using chiral plasmonic NPs in CP-OLED architectures has not been previously reported. Moreover, we have found, for the first time, that chiral NPs can trigger the production of high-quality chiral excitons for circularly polarized photo- and electroluminescence. We therefore believe the achievement of CP-OLEDs utilizing chiral plasmonic NPs is of high novelty, as is also endorsed by Reviewers #1, #2, and #3.

Reviewer's statement (b): *There are publications already showing EQE around 2.5% and g_{EL} factor approximately around 0.32 utilizing different chiral constructs* (<https://www.nature.com/articles/s41467-022-35699-z>, <https://www.chinesechemsoc.org/doi/full/10.31635/ccschem.023.202303346>). *So in terms of EQE and g_{EL} , this is not a substantial improvement.*

Response: We have read the 2 reference publications suggested by the reviewer carefully. The first publication shows enhanced circularly polarized photoluminescence from plasmonic nanorods decorated with chiral dye-loaded micelles, with the highest photoluminescence dissymmetry factor of 0.062. We could not find any data on the fabrication of CP-OLED devices or the EQE and g_{EL} factor data of electroluminescence in this work. The second one is a review article, which introduces the development of CP-OLEDs based on thermally activated delayed fluorescence and phosphorescent materials. We really appreciate that the reviewer provided such a nice reference. However, we could not find any statements showing the EQE around 2.5% and g_{EL} factor around 0.32 in the review article. Besides, we have summarized the EQE values and g_{EL} factor of our work and compared with those of CP-OLED devices introduced in the review article in our revised manuscript. Many CP-OLEDs shown in this review article have large EQEs (>10%) but small g_{EL} factors (<0.01). This review article therefore cannot support the statement (b). Conversely, it proves that our CP-OLED devices have achieved a substantial improvement in the CPEL performance (see the figure below).

Caption | Comparison of various CP-OLEDs in terms of EQE and $|g_{EL}|$.

Overall, we greatly appreciate the reviewer's great effort in reviewing our manuscript and providing suggestions that aim to improve our work. However, we are afraid that we can hardly agree with the reviewer's statement on the novelty and device performance. From the references provided by the reviewer, we could not find the information that the reviewer mentioned in her/his comments. But these references are still helpful and we have added the related discussion accordingly in our revised manuscript. We sincerely hope that the innovation and soundness of our work can be re-evaluated.

Question 1: In figure 1(d, e) and related supporting information figure(s), the influence of Chiral plasmon & chiral plasmon-chiral exciton has been discussed. However, there is no detailed optical modes (dark/bright) discussion for such standalone plasmonic structures versus Plasmonic + TBDC structures.

Response: We thank the reviewer for this interesting question. In our study, we characterized the chiral plasmon–chiral exciton interaction by extinction CD spectra describing different extinction for LCP and RCP incident light. It is challenging to employ the CD spectra, i.e., the differential extinction spectra, to analyze information of dark and bright optical modes. Absorption/extinction and scattering measurements on the single-particle level are usually required to characterize optical modes, as demonstrated in previous studies (*Phys. Rev. Res.* **2020**, 2, 033056; *Nano Lett.* **2022**, 22, 4686). In our work, we met some difficulties of characterizing the single-nanostructure absorption/extinction and scattering properties, since our samples suffered from strong excessive background noise originating from TBDC not coupled with the plasmonic NPs (see the figure below). Our study involves various chiral plasmonic NPs and a significant amount of experimental work should be done to clearly disclose the optical modes and their detailed interactions in the hybrid system of the chiral plasmonic NP and TBDC aggregates supporting chiral excitons. Despite the lack of such study, we offered a theoretical model for explaining the optical responses of the chiral plasmonic–excitonic systems in Supplementary Information. More importantly, the main findings of this manuscript on circularly polarized luminescence and CP-OLED devices will not be affected. We are planning to conduct further detailed investigations on the optical modes and their complex interactions in the hybrid system containing chiral plasmons and chiral excitons.

Caption | Extinction and CD spectra of the 432 helicoid III–TBDC hybrids. It is difficult to observe the optical dark/bright modes owing to strong excessive background noise originating from TBDC absorption peaks.

Question 2: How about the optical properties involving tuning/detuning of upper/lower excitons for a chiral plasmonic NPs versus TBDC coupled chiral plasmonic NPs (differences in term of plasmon & exciton).

Response: We find some difficulty in understanding Question 2 of the reviewer. We guess that the reviewer would like to ask about changes in the optical properties of the nanostructures before and after the chiral plasmonic NPs are coupled with TBDC. We have

added some discussion on the chiroptical properties of the TDBC-coupled chiral plasmonic NPs. In detail, we synthesized the 432 helicoid III NPs with tunable CD peaks ranging from 540 to 600 nm. TDBC aggregates were then assembled from the 432 helicoid III NPs with varying chiroptical properties. We first measured the extinction CD spectra of the (432 helicoid III NP)–(TDBC aggregate) hybrids before and after the evaporation of water, as shown in Supplementary Fig. 9. Previous works have explored strong chiral light–matter interactions in a TDBC solution mixed with chiral plasmonic NPs and nanostructures (*Nano Lett.* **2023**, *23*, 11376; *Nano Lett.* **2021**, *21*, 3573). Similar to the previous reports, detuning between chiral plasmons and excitons can strongly influence the CD response of the (432 helicoid III NP)–(TDBC aggregate) hybrid solution (Supplementary Fig. 9). The CD spectra exhibited splitting, which showed an upper branch mode and a lower branch mode. Both of the modes shift as the plasmonic CD peak of the chiral NPs is varied. After the evaporation of water, we prepared the (432 helicoid III NP)–(TDBC aggregate) hybrid film. More chiral excitons of TDBC aggregates are formed on the chiral NP surface because of the accumulation of TDBC molecules. The chiral excitons have a chiroptical response around the extinction band of TDBC (585 nm). We then observed enlarged splitting in the CD spectra when the hybrid system was changed from aqueous solution to solid film.

We have revised Supplementary Fig. 9 and provided the discussion about the chiroptical properties in the caption.

Supplementary Fig. 9 | Extinction CD spectra of the (432 helicoid III NP)-(TDBC aggregate) hybrids. a,b SEM images (**a**) and extinction CD spectra (**b**) of the L-432 helicoid III NPs. **c,d** Extinction CD spectra of the (L-432 helicoid III NP)-(TDBC aggregate) hybrids before (**c**) and after (**d**) the evaporation of water during the preparation of the hybrid films. **e,f** SEM images (**e**) and extinction CD spectra (**f**) of the D-432 helicoid III NPs. **g,h** Extinction CD spectra of the (D-432 helicoid III NP)-(TDBC aggregate) hybrids before (**g**) and after (**h**) the evaporation of water during the preparation of the hybrid films. The detuning between chiral plasmons and excitons can strongly affect the CD response of the (432 helicoid III NP)-(TDBC aggregate) hybrid solution, resulting in the splitting between an upper plexciton branch mode and a lower plexciton branch mode. After the evaporation of water, (432 helicoid III NP)-(TDBC aggregate) hybrid films were fabricated, causing more chiral excitons in the TDBC aggregates formed on the chiral NP surface. The chiral excitons have a chiroptical response around the extinction band of TDBC (585 nm). We then observed the enlarged splitting in the CD spectra when the hybrid system was changed from the aqueous solution to the solid film.

Question 3: As author discuss about the significance of plasmonic enhancement as one of the important parameter in achieving high EQE and g_{EL} , the optical background study directly influencing EQE (fig 4b) is not clear! Discussion on how the near-field properties fare better in their plasmonic constructs are needed!

Response: We thank this reviewer for the insightful suggestion. Previous works (*Nature* **2020**, 585, 379; *Nat. Commun.* **2023**, 14, 81) have demonstrated that plasmonic NPs support strong enhancement of electromagnetic field at the NP surface, which can significantly increase the PL and EL intensity. We calculated the electric field intensity enhancement around the 432 helicoid III NP by FDTD simulation and found that the plasmonic enhancement $|E|^2/|E_0|^2$ reaches 40 at the tips of the 432 helicoid III NP at a wavelength close to the emission wavelength of the TDBC molecule (Supplementary Fig. 34 in the updated manuscript, please also see the figure below). The large plasmonic near-field leads to the plasmonic Purcell effect (please refer to Reference 34 in the revised manuscript). Excitons near the NP surface therefore decay radiatively in a faster manner. The accelerated radiative process can result in an enlarged emission quantum efficiency, and therefore give rise to a higher EQE.

We have added the calculation results as Supplementary Fig. 34 and provided the discussion about plasmonic enhancement in the caption.

Supplementary Fig. 34 | FDTD-calculated plasmonic near-field enhancement $|E|^2/|E_0|^2$ of a chiral Au 432 helicoid III NP. **a** Schematic showing the 432 helicoid III NP (top) and simulation-calculated absorption and scattering spectra of the 432 helicoid III NP under the excitation of circularly polarized light. A plasmonic peak at ~600 nm was observed from the calculated spectra, right at the CPEL emission peak of the CP-OLEDs. **b** Calculated distribution of plasmonic electric field enhancement $|E|^2/|E_0|^2$ at the surface of the Au 432 helicoid III NP when the excitation wavelength was 600 nm. The maximum plasmonic enhancement was found to be 40, located at the tips of the 432 helicoid III NP. The calculated results suggested that the plasmonic near-field enhancement from 432 helicoid III might be another reason for high EQEs of the CP-OLED-2 devices. The large plasmonic near-field of the 432 helicoid III NP can lead to a significant plasmonic Purcell effect. Excitons near the NP surface therefore decay radiatively in a faster manner. The accelerated radiative process

can result in an enlarged emission quantum efficiency, and therefore give rise to higher EQEs of the chiral plasmonic NP-based CP-OLED devices.

We have also provided the discussion about plasmonic enhancement in the first paragraph under the “Luminous efficiency of the CP-OLEDs” section in the main text.

“..... The superior EQE performance of the CP-OLED-2 devices can be attributed to the following two reasons. The first is the large plasmonic near-field enhancement near the surface of the 432 helicoid III NP (Supplementary Fig. 34). The plasmonic near-field enhancement can accelerate the radiation process of excitons because of the Purcell effect³⁴, giving rise to a higher emission quantum efficiency. The plasmonic NPs further work as antennas that can increase the external extraction efficiency^{44,45}. The second reason is the suppression of the overshoot effect that was observed in the Ref-OLED and CP-OLED-3 devices. The overshoot effect came from

The references (*Nature* **2020**, 585, 379; *Nat. Commun.* **2023**, 14, 81) have been added.

44. Fusella, M. A. *et al.* Plasmonic enhancement of stability and brightness in organic light-emitting devices. *Nature* **585**, 379–382 (2020).
45. Zhao, T. H. *et al.* Enhanced chiroptic properties of nanocomposites of achiral plasmonic nanoparticles decorated with chiral dye-loaded micelles. *Nat. Commun.* **14**, 81 (2023).

Question 4: *Supp. Fig. 21 color scale shows only +/- distribution which can't be a parameter to estimate the plasmonic enhancement. It can reveal the plasmonic modes but not the enhancement factor.*

Response: Supplementary Fig. 17 (Supplementary Fig. 21 in the original version) shows the FDTD-calculated optical chirality enhancement $C/|C_{CPL}|$ of the chiral Au 432 helicoid III NP and nanotriskelion. We had provided the enhancement factor in number rather than only in signs. In detail, we calculated the parameter $C/|C_{CPL}|$ to characterize the near-field chirality distribution of the chiral NPs. The value of $C/|C_{CPL}|$ varies from -1 for perfectly LCP light to $+1$ for perfectly RCP light. A chiral nanostructure with $C/|C_{CPL}| > 1$ is considered to support super-chiral near-field, implying the electromagnetic near-field distribution shows a greater optical chirality than a perfectly circularly polarized plane wave. Previous works have used the optical chirality enhancement to demonstrate that 2D-chiral perovskite nanocrystal metasurfaces can transfer optical chirality to unpolarized light and trigger CPL emissions (*Adv. Mater.* **2023**, 35, 2210477). Similarly, we demonstrate that the chiral Au 432 helicoid III NP and nanotriskelion provide a strong chiral optical near-field that can modulate CPPL and CPEL.

Question 5: *For the overshoot effect based results, how about the recovery time (for a multiple continuous on/off cases, as seen from Supp. Fig. 27b, d). Furthermore, as seen from the time-dependent EL, the tail signature remains ~ 0.3 seconds! In such situation, how about a device facing continuous on-off state and recovery? How it can affect the life-time stability such CP-OLED device?*

Response: We thank the reviewer for the valuable questions. A detailed explanation on the overshoot effect and how the overshoot effect affects the CPEL performance of our CP-OLEDs have been added in the main text (please refer to our response to Question 2 raised by Reviewer #1 above). We have further addressed the questions one by one as follows.

(1) We have provided detailed explanation about the overshoot effect. The experimental results about the overshoot effect are summarized in Supplementary Fig. 22 in the revised manuscript. Previous works have proved that upon the application of a pseudo-rectangular voltage pulse to a dopant-emitting OLED, the luminance can increase and overshoot to a maximum value before decreasing to a steady value (*Chem. Phys. Lett.* **2004**, 397, 87; *J. Lumin.* **2022**, 246, 118850). TDBC monomers have two imidazole rings and show electron-donating properties. In the Ref-OLED device, after a few rounds of electrical switching, some electrons were pre-trapped in the TDBC aggregates. The new round of electrical switching triggered the radiative recombination of injected holes and the pre-stored electrons, producing a spike with a super-high intensity at the transient EL rising edge. As shown in Supplementary Fig. 22c, a recovery time of 0.1–0.2 s is necessary for the EL of Ref-OLED to reach its steady state. As a comparison, we also measured the transient EL spectra and recorded the images of the transient luminescence process for the CP-OLED-2 devices. The CP-OLED-1 and CP-OLED-2 devices used the hybrids of TDBC aggregates and the chiral Au NPs with excellent electrical conductivity. The amount of pre-stored electrons can be significantly reduced, thereby eliminating the overshoot effect. Compared with Ref-OLED, the luminance of CP-OLED-2 underwent a relatively slow rise time (about 0.2 s) to reach a steady state. This is because of the emissive layer with a large thickness of 640 ± 50 nm. After the application of a voltage, the transport and recombination of charge carriers required a longer time to reach the steady state.

We have revised Supplementary Fig. 22 and added the detailed explanation about the overshoot effect and the recovery time in the caption of Supplementary Fig. 22.

Supplementary Fig. 22 | Comparison of the Ref-OLED and CP-OLED-2 devices in terms of the overshoot effect. **a** Time-dependent luminating photographs of a Ref-OLED device and a CP-OLED-2 device after a voltage of $U = 6$ V was applied. **b** Normalized EL intensity changes over time for a Ref-OLED device and a CP-OLED-2 device under the application of $U = 4$ V. The CP-OLED-2 device with metal NPs exhibits a higher stability because of the suppression of the overshoot effect. **c** Transient EL profiles of a Ref-OLED device under the application of $U = 4$ V. **d** Schematic illustrating the overshoot phenomenon in the Ref-OLED and CP-OLED-3 devices. The luminance first increases and overshoots to a maximum value (i) before decreasing to a steady value (ii). TDBC monomers have two imidazole rings (Fig. 1c) and show electron-donating properties. In the Ref-OLED device, after a few rounds of electrical switching, some electrons were pre-stored in the TDBC aggregates. The new round of electrical switching triggers the radiative recombination of the injected holes and the pre-stored electrons (i), producing a spike with a super-high intensity at the transient EL rising edge⁷. After the pre-stored electrons are consumed because of recombination with injected holes, the luminance decreases until it reaches a steady value, and new electrons start to accumulate in the TDBC aggregates again (ii). A recovery time of 0.1–0.2 s is necessary for the EL of Ref-OLED to reach its steady state. The overshoot effect can also be observed in the CP-OLED-3 devices using separate TDBC and Au NP layers. **e** Transient EL profile of a CP-OLED-2 device under the application of $U = 4$ V. **f** Schematic

illustrating the suppression of the overshoot effect in the CP-OLED-1 and CP-OLED-2 devices. The CP-OLED-1 and CP-OLED-2 devices used the hybrids of TDBC aggregates and the chiral Au NPs with excellent electrical conductivity. The amount of pre-stored electrons can be significantly reduced, thereby eliminating the overshoot effect. Compared with Ref-OLED, the measured CP-OLED-2 device underwent a relatively slow rise in its luminance intensity upon the application of a voltage, which took about 0.2 s until the luminance intensity reached a steady state. The slower rise of the luminance intensity in the CP-OLED-2 devices compared with that of their Ref-OLED counterparts results from both the lack of pre-stored carriers and the larger emissive layer thickness of the CP-OLED-2 devices (640 ± 50 nm *versus* 560 ± 37 nm). After the application of a voltage pulse, the transport and recombination of charge carriers requires longer time.

(2) We have provided additional detailed experimental results for the OLED devices undergoing continuous on/off states. The transient EL data of the Ref-OLED and CP-OLED-3 devices under 4 cycles of on/off operation are shown in Supplementary Figs. 21 in the revised manuscript. Supplementary Figs. 28–33 further show the transient EL spectra of different CP-OLED-2 devices under various cycles of on/off operation at different voltages. The OLED devices can work properly under continuous on/off states.

(3) For the lifetime and stability of the CP-OLED devices, we have added detailed experimental results on examining the stability of the Ref-OLED and CP-OLED-2 devices in Supplementary Figs. 22b. Because of the severe accumulation of electrons and holes inside the emissive layer, which may degrade the chromophore molecules, the luminous lifespan of Ref-OLED was shorter than that of CP-OLED-2 devices.

We have added the experimental results on the stability of the Ref-OLED and CP-OLED-2 devices in revised Supplementary Figs. 22b.

Supplementary Fig. 22 | Comparison of the Ref-OLED and CP-OLED-2 devices in terms of the overshoot effect. b Normalized EL intensity changes over time for a Ref-OLED device and a CP-OLED-2 device under the application of $U = 4$ V. The CP-OLED-2 device with metal NPs exhibits a higher stability because of the suppression of the overshoot effect.

We have also added the discussion on how the overshoot effect affects the lifetime and stability of the OLED devices working under continuous on/off states at the end of the paragraph right before Figure 5 in the main text.

“..... The overshoot effect came from the radiative recombination of pre-stored electrons and injected holes, characterized by a spike with a super-high intensity at the transient EL rising edge (Supplementary Fig. 22)^{46,47}. We found that the overshoot effect can make the OLEDs work in an unstable state, reducing the EQE and luminous lifespan (Supplementary Fig. 22b and Fig. 23). The transient EL examination of the CP-OLED-2 devices have revealed that the chiral NPs with excellent electrical conductivity is helpful to significantly reduce the amount of pre-stored electrons, eliminating the overshoot effect (Supplementary Fig. 22).”

Response to Reviewer #5

Comments: I co-reviewed this manuscript with one of the reviewers who provided the listed reports. This is part of the Nature Communications initiative to facilitate training in peer review and to provide appropriate recognition for Early Career Researchers who co-review manuscripts.

Response: We thank this reviewer for the great effort on evaluating our work and the insightful inputs! They really helped a lot in improving the quality of our manuscript.

REVIEWERS' COMMENTS

Response to Reviewer #1

Comments: After careful evaluation of revised manuscript, I believe that the manuscript can be suitable for publication in journal "Nature Communications" in presented form. All necessary additions and clarifications are made in the revised manuscript.

Response: We thank this reviewer for the effort on evaluating our work again, the highly positive comment, and the great help in the improvement of our manuscript.

Response to Reviewer #2

Comments: As far as I am concerned, the manuscript may be published in the present form, since the Authors have correctly addressed all the issues I have raised.

Response: We thank this reviewer for the effort on evaluating our work again, the highly positive comment, and the great help in the improvement of the manuscript.

Response to Reviewer #3

Comments: I have carefully considered the revisions made by the authors. The manuscript has been improved and the critical aspects have been addressed by the authors. For this reason, I think that the manuscript can now be accepted in Nat. Comm..

Response: We thank this reviewer for the effort on evaluating our work again, the highly positive comment, and the great help in the improvement of the manuscript.

Response to Reviewer #5

Comments: I co-reviewed this manuscript with one of the reviewers who provided the listed reports. This is part of the Nature Communications initiative to facilitate training in peer review and to provide appropriate recognition for Early Career Researchers who co-review manuscripts.

Response: We thank this reviewer for the effort on evaluating our work again, the highly positive comment, and the great help in the improvement of the manuscript.